# Machine learning-based glucose prediction with use of continuous glucose and physical activity monitoring data: The Maastricht Study

**William P. T. M. van Doorn**[1,2�], **Yuri D. Foreman**[1,3�], **Nicolaas C. Schaper**[1,4,5], **Hans H. C. M. Savelberg**[6], **Annemarie Koster**[5,7], **Carla J. H. van der Kallen**[1,3], **Anke Wesselius**[8], **Miranda T. Schram**[1,3,9], **Ronald M. A. Henry**[1,3,9], **Pieter C. Dagnelie**[1,3], **Bastiaan E. de Galan**[1,4,10], **Otto Bekers**[1,2], **Coen D. A. Stehouwer**[1,3], **Steven J. R. Meex**[1,2‡], **Martijn C. G. J. Brouwers**[1,4‡]*

1 CARIM School for Cardiovascular Diseases, Maastricht University, Maastricht, The Netherlands, 2 Department of Clinical Chemistry, Central Diagnostic Laboratory, Maastricht University Medical Centre+, Maastricht, The Netherlands, 3 Department of Internal Medicine, Maastricht University Medical Centre+, Maastricht, The Netherlands, 4 Division of Endocrinology and Metabolic Disease, Department of Internal Medicine, Maastricht University Medical Centre+, Maastricht, The Netherlands, 5 CAPHRI Care and Public Health Research Institute, Maastricht University, Maastricht, The Netherlands, 6 Department of Human Biology and Movement Science, NUTRIM School for Nutrition and Translational Research in Metabolism, Maastricht University, Maastricht, The Netherlands, 7 Department of Social Medicine, Maastricht University, Maastricht, The Netherlands, 8 Department of Complex Genetics and Epidemiology, NUTRIM School for Nutrition and Translational Research in Metabolism, Maastricht University, Maastricht, The Netherlands, 9 Heart and Vascular Centre, Maastricht University Medical Centre+, Maastricht, The Netherlands, 10 Department of Internal Medicine, Radboud University Medical Centre, Nijmegen, The Netherlands

y These authors contributed equally to this work.
‡ These authors also contributed equally to this work.
* mcgj.brouwers@mumc.nl

**Data Availability Statement:** Data are unsuitable for public deposition due to ethical restriction and privacy of participant data. The study has been approved by the medical ethical committee of the

## Abstract

### Background

Closed-loop insulin delivery systems, which integrate continuous glucose monitoring (CGM) and algorithms that continuously guide insulin dosing, have been shown to improve glycaemic control. The ability to predict future glucose values can further optimize such devices. In this study, we used machine learning to train models in predicting future glucose levels based on prior CGM and accelerometry data.

### Methods

We used data from The Maastricht Study, an observational population-based cohort that comprises individuals with normal glucose metabolism, prediabetes, or type 2 diabetes. We included individuals who underwent >48h of CGM (n = 851), most of whom (n = 540) simultaneously wore an accelerometer to assess physical activity. A random subset of individuals was used to train models in predicting glucose levels at 15- and 60-minute intervals based on either CGM data or both CGM and accelerometer data. In the remaining individuals, model performance was evaluated with root-mean-square error (RMSE), Spearman's correlation coefficient (rho) and surveillance error grid. For a proof-of-concept translation, CGM-

Maastricht University Medical Center (NL31329.068.10/ MEC 10-2-023) and the Netherlands Health Council under the Dutch "Law for Population Studies" (Permit 131088-105234-PG). Data are available from The Maastricht Study for any interested researchers who meet the criteria for access to confidential data. The Maastricht Study Management Team (research. dms@mumc.nl) and the corresponding author (Martijn C.G.J. Brouwers) may be contacted to request data.

**Funding:** The Maastricht Study was supported by the European Regional Development Fund via OP-Zuid, the Province of Limburg, the Dutch Ministry of Economic Affairs (grant 310.041), Stichting De Weijerhorst (Maastricht, the Netherlands), the Pearl String Initiative Diabetes (Amsterdam, the Netherlands), School for Cardiovascular Diseases (CARIM, Maastricht, the Netherlands), School for Public Health and Primary Care (CAPHRI, Maastricht, the Netherlands), School for Nutrition and Translational Research in Metabolism (NUTRIM, Maastricht, the Netherlands), Stichting Annadal (Maastricht, the Netherlands), Health Foundation Limburg (Maastricht, the Netherlands), and by unrestricted grants from Janssen-Cilag B.V. (Tilburg, the Netherlands), Novo Nordisk Farma B. V. (Alphen aan den Rijn, the Netherlands), Sanofi-Aventis Netherlands B.V. (Gouda, the Netherlands), and Medtronic (Tolochenaz, Switzerland). The funders had no role in study design, data collection and analysis, decision to publish, or preparation of the manuscript.

**Competing interests:** This study received funding from Janssen-Cilag B.V, Novo Nordisk Farma B.V., Sanofi-Aventis Netherlands B.V and Medtronic. There are no patents, products in development or marketed products to declare. This does not alter the authors' adherence to all the PLOS ONE policies on sharing data and materials, as detailed online in the guide for authors. N.C. Schaper, R.M. A. Henry, and M.C.G.J. Brouwers were supported by Medtronic (External Research Program). Medtronic did not direct design, conduct, or outcomes of this study. The authors declare that there are no other relationships or activities that might bias, or be perceived to bias, their work.

based prediction models were optimized and validated with the use of data from individuals with type 1 diabetes (OhioT1DM Dataset, n = 6).

## Results

Models trained with CGM data were able to accurately predict glucose values at 15 (RMSE: 0.19mmol/L; rho: 0.96) and 60 minutes (RMSE: 0.59mmol/L, rho: 0.72). Model performance was comparable in individuals with type 2 diabetes. Incorporation of accelerometer data only slightly improved prediction. The error grid results indicated that model predictions were clinically safe (15 min: >99%, 60 min >98%). Our prediction models translated well to individuals with type 1 diabetes, which is reflected by high accuracy (RMSEs for 15 and 60 minutes of 0.43 and 1.73 mmol/L, respectively) and clinical safety (15 min: >99%, 60 min: >91%).

## Conclusions

Machine learning-based models are able to accurately and safely predict glucose values at 15- and 60-minute intervals based on CGM data only. Future research should further optimize the models for implementation in closed-loop insulin delivery systems.

## Introduction

The increasing prevalence of diabetes entails an increase in debilitating complications, such as retinopathy, neuropathy, and cardiovascular disease [1–3]. Maintaining plasma glucose levels within the reference range is essential for the prevention of diabetes-related complications, which are generally attributable to chronic hyperglycaemia, although hypoglycaemia has been suggested to contribute to cardiovascular disease risk as well [3–5]. One of the most promising developments to minimize hyperglycaemia and hypoglycaemia–and, hence, to increase time in range–in individuals with diabetes who require insulin treatment is a closed-loop insulin delivery system (also known as the artificial pancreas). Such a system integrates continuous glucose monitoring (CGM), insulin (with or without glucagon) infusion, and a control algorithm to continuously regulate blood glucose levels [6, 7]. Multiple studies have shown the merit of incorporating the artificial pancreas into clinical care of individuals with type 1 or type 2 diabetes [8, 9].

Despite prior efforts, there are still numerous points that need to be addressed in order to improve the individual components of closed-loop systems [6, 10]. With regard to CGM, this includes overcoming sensor delay (i.e., the inherent ~10-minute discrepancy between interstitially measured and actual plasma glucose values), and sensor malfunctions (i.e., periods during which no glucose values are recorded) [6, 10, 11]. Continuous glucose prediction is a potentially viable strategy to both handle sensor delay and bridge periods of sensor malfunction. The use of machine learning has yielded encouraging glucose prediction accuracy results in relatively small study populations (mostly individuals with type 1 diabetes) or *in silico* studies, as extensively reviewed elsewhere [12]. Large, human-based study populations are now needed to reliably assess to what extent and within what time interval (i.e., prediction horizon) glucose values can be accurately predicted by use of machine learning. Additionally, incorporation of physical activity, which is considered an important factor for glucose control in daily life, could further improve glucose prediction [6].

In this study, we investigated to what extent glucose values can be accurately predicted at intervals of 15 and 60 minutes by a machine learning model that has been trained with a sliding time window of glucose values preceding the predicted values at a fixed interval. Additionally, we studied whether glucose prediction can be further improved by incorporation of accelerometer-measured physical activity, and to what extent the results differ in a subgroup analysis of individuals with type 2 diabetes only. For this, we used a large population of individuals with either normal glucose metabolism (NGM), prediabetes, or type 2 diabetes who simultaneously underwent CGM and continuous accelerometry during a one-week period. Last, we used the publicly available OhioT1DM Dataset to explore whether CGM-based prediction models would translate to individuals with type 1 diabetes, the primary target population for closed-loop insulin delivery.

## Methods

### Study population and design

We used data from The Maastricht Study, an observational, prospective, population-based cohort study. The rationale and methodology have been described previously [13]. In brief, The Maastricht Study focuses on the aetiology, pathophysiology, complications and comorbidities of type 2 diabetes, and is characterized by an extensive phenotyping approach. All individuals aged between 40 and 75 years and living in the southern part of the Netherlands were eligible for participation. Participants were recruited through mass media campaigns and from the municipal registries and the regional Diabetes Patient Registry via mailings. For reasons of efficiency, recruitment was stratified according to known type 2 diabetes status, with an oversampling of individuals with type 2 diabetes. In general, the examinations of each participant were performed within a time window of three months. From 19 September 2016 until 13 September 2018, participants were invited to also undergo CGM [14]. During this period, a selected group of recently included participants were invited to return for CGM. In these participants only, there was a median time interval of 2.1 years between CGM and all other measurements. The present report includes cross-sectional data of the 851 participants who had at least 48h of CGM data available and were classified with NGM, prediabetes, or type 2 diabetes. The Maastricht Study has been approved by the institutional medical ethical committee (Medisch-ethische toetsingscommissie aZM/UM [METC]; NL31329.068.10) and the Minister of Health, Welfare and Sports of the Netherlands (Permit 131088-105234-PG). All participants gave written informed consent.

### Continuous glucose monitoring

The rationale and methodology of CGM (iPro2 and Enlite Glucose Sensor; Medtronic, Tolochenaz, Switzerland) have been described previously [14]. In brief, the CGM device was worn abdominally and recorded subcutaneous interstitial glucose values (range: 2.2–22.2 mmol/L) every five minutes for a seven-day period. For calibration purposes, participants were asked to perform self-measurements of blood glucose four times daily (Contour Next; Ascensia Diabetes Care, Mijdrecht, the Netherlands). Participants were blinded to the CGM recording, but not to self-measured values. Diabetes medication use was allowed and no dietary instructions were given. We only included individuals with at least 48h of CGM, but excluded the first 24h of CGM from analysis because of insufficient calibration. For the glucose prediction analyses, all remaining glucose data points were used. We additionally calculated mean sensor glucose, standard deviation (SD), and coefficient of variation (CV) with the use of Glycemic Variability Research Tool (GlyVaRT; Medtronic) software.

## Accelerometry

As described previously, daily physical activity was measured with use of the triaxial activPAL3 accelerometer (PAL technologies; Glasgow, United Kingdom) [13, 15]. The accelerometer was, just as the CGM device, attached during the first research visit; participants wore the accelerometer on the front of the right thigh for eight consecutive days. No physical activity instructions were given. PAL Software Suite version 8 (PAL technologies) was used to convert the event-based accelerometry data files into 15-second interval data files. We used the composite of X, Y, and Z accelerations for each 15-second interval as the measure of physical activity.

## Assessment of participant characteristics

As described previously [13], we classified glucose metabolism status (GMS) as either NGM, prediabetes, or type 2 diabetes based on both a standardized 2-hour 75 gram oral glucose tolerance test and use of glucose-lowering medication [16]. We assessed medication use as part of a medication interview. Additionally, we determined smoking status and history of diabetes based on questionnaires, measured weight and height–to calculate body mass index (BMI)–and office blood pressure during a physical examination, and measured $HbA_{1c}$ as well as lipid profile in fasting venous blood.

## Dataset construction

An overview of data preprocessing, model development, and model evaluation is given in Fig 1. In order to train our models in predicting future glucose values, we constructed two separate datasets (Fig 1, panel a). The first dataset consisted of only the participants' six-day, five-minute interval CGM data (n = 851). The second dataset consisted of both CGM and accelerometry data (n = 540). To synchronize CGM (determined at 5-minute intervals) and accelerometry data (determined at 15-second intervals) in the second dataset, we linearly interpolated glucose values between two glucose data points with a frequency of 15 seconds. Consistent and aligned frequency intervals across these parameters are a statistical precondition for this type of model development [17]. The study populations were randomly split into a training (70%), tuning (10%), and evaluation (20%) dataset such that data from a given individual were present only in one set. The training set was used to train the proposed models. The tuning set was used to iteratively improve the models by selecting the best model architectures and hyperparameters. Finally, the best models were evaluated on the independent evaluation set that was retained during model development.

## Model development and design

Our proposed predictive model operates sequentially over CGM and accelerometry data (Fig 1, panel b). At each individual time point, 30 minutes of prior time series data were provided to the statistical model (e.g., six CGM-based glucose values), based on which it predicted glucose values at specified time intervals. For this study, we set these time intervals at 15 and 60 minutes. The nature of this prediction task can be solved by a variety of statistical and machine learning models. In the current study, we assessed autoregressive integrated moving average, support vector regression, gradient-boosting systems, shallow and deep multi-layer perceptron neural networks, and several recurrent neural network (RNN) architectures, including classical RNN [18, 19], gated recurrent units [20], long-short term memory (LSTM) networks [21], and all of its bi-directional variants [22, 23] (S1 File).

**a,** Data preprocessing

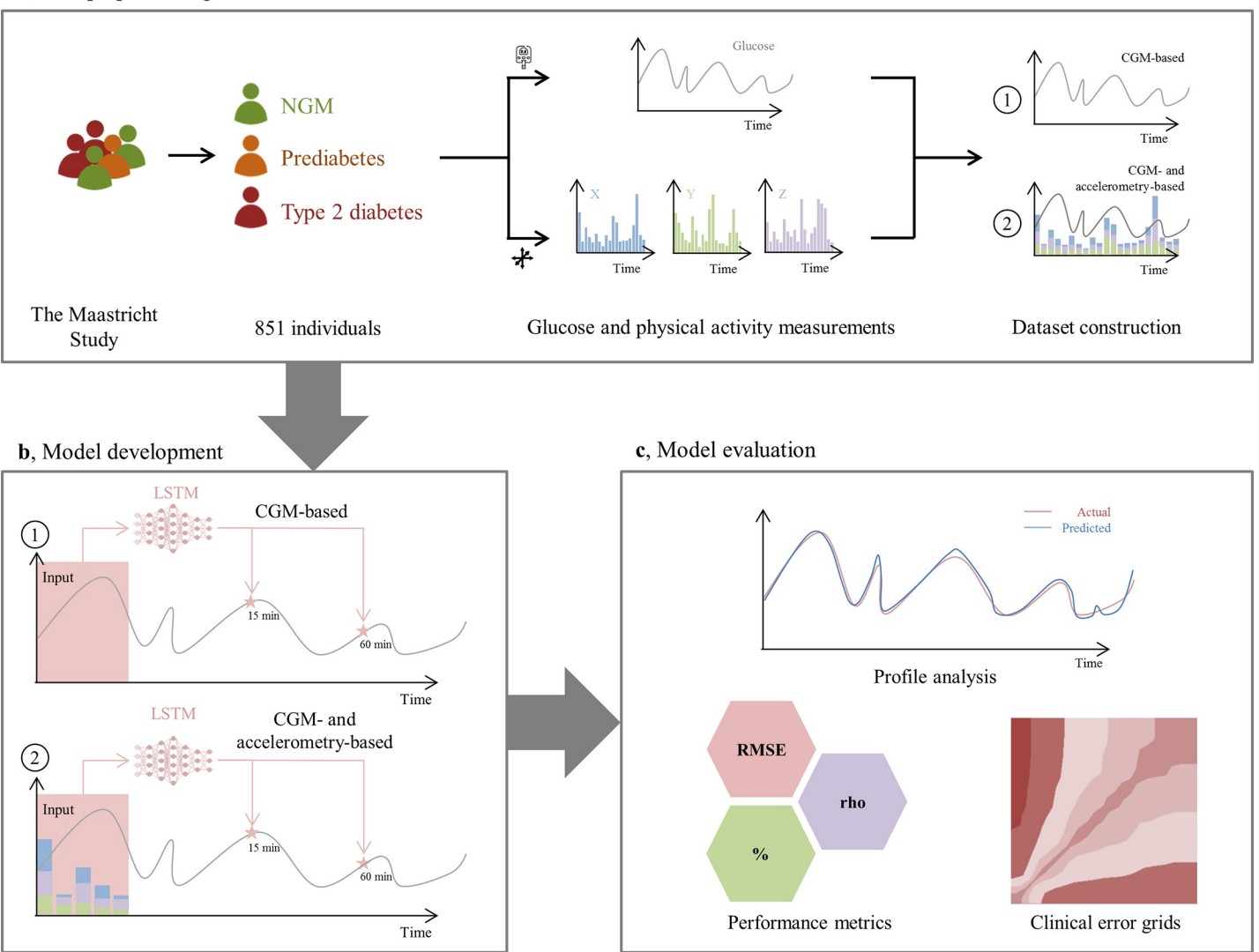

**Fig 1. Overview of data preprocessing, model development and evaluation.** Data was used from The Maastricht Study, an observational population-based cohort that comprises individuals with normal glucose metabolism (NGM), prediabetes, or type 2 diabetes (panel A). We included 851 individuals who underwent continuous glucose monitoring (CGM), most of whom simultaneously wore an accelerometer to assess physical activity (X, Y, and Z accelerations). Models developed with the long-short term memory (LSTM) architecture were trained in predicting glucose levels at 15- and 60-minute intervals with either CGM data only (1) or both CGM and accelerometer data (2) (panel B). Finally, model performance was evaluated by glucose profile analysis, performance metrics (root-mean-square error [RMSE]; Spearman's correlation coefficient [rho]; proportions), and clinical error grids (panel C).

## Model selection and training

The classical RNN architecture had superior performance at the 15-minute prediction interval (Table 1, RMSE: 0.485 [0.481–0.490]), whilst the LSTM network outperformed all other architectures at the 60-minute prediction interval (Table 1, RMSE: 0.941 [0.937–0.945]). Considering the performance of the LSTM network at a 15-minute prediction interval was nearly as good as the classical RNN, we selected the multi-task LSTM network among several alternatives as architecture of choice to continue our investigations(S1 File and Table 1). This architecture runs sequentially over time series data and is able to implicitly model the historical context of an individual by modifying an internal state through time. Specifically, we designed

**Table 1. Baseline statistical and machine learning model comparison for predicting glucose values.**

| Prediction window and baseline model | | CGM-based glucose prediction | | Combined glucose prediction | |
|---|---|---|---|---|---|
| | | Rho | RMSE, mmol/L | Rho | RMSE, mmol/L |
| **15 minutes** | ARIMA | 0.842 [0.837–0.848] | 0.504 [0.490–0.518] | 0.834 [0.829–0.840] | 0.498 [0.492–0.505] |
| | SVR | 0.791 [0.781–0.802] | 0.558 [0.549–0.567] | 0.703 [0.694–0.712] | 0.612 [0.601–0.622] |
| | LightGBM | 0.783 [0.767–0.795] | 0.589 [0.577–0.601] | 0.783 [0.771–0.794] | 0.497 [0.582–0.613] |
| | Shallow MLP | 0.810 [0.804–0.816] | 0.517 [0.506–0.529] | 0.763 [0.754–0.772] | 0.592 [0.581–0.603] |
| | Deep MLP | 0.807 [0.797–0.818] | 0.511 [0.504–0.518] | 0.828 [0.819–0.837] | 0.510 [0.503–0.517] |
| | **RNN** | **0.894 [0.887–0.902]** | **0.485 [0.481–0.490]** | **0.890 [0.882–0.898]** | **0.477 [0.472–0.482]** |
| | LSTM | 0.872 [0.865–0.879] | 0.482 [0.477–0.487] | 0.884 [0.878–0.890] | 0.501 [0.496–0.506] |
| **60 minutes** | ARIMA | 0.307 [0.284–0.329] | 1.543 [1.489–1.623] | 0.303 [0.283–0.322] | 1.502 [1.455–1.568] |
| | SVR | 0.388 [0.376–0.398] | 1.386 [1.322–1.452] | 0.394 [0.382–0.405] | 1.412 [1.350–1.475] |
| | LightGBM | 0.500 [0.491–0.508] | 1.118 [1.098–1.136] | 0.498 [0.485–0.511] | 1.128 [1.107–1.148] |
| | Shallow MLP | 0.503 [0.495–0.511] | 1.081 [1.074–1.088] | 0.483 [0.470–0.495] | 1.081 [1.070–1.092] |
| | Deep MLP | 0.496 [0.484–0.509] | 1.108 [1.100–1.115] | 0.515 [0.502–0.528] | 1.108 [1.099–1.017] |
| | RNN | 0.591 [0.581–0.600] | 0.989 [0.983–0.995] | 0.596 [0.589–0.603] | 0.992 [0.984–0.998] |
| | **LSTM** | **0.605 [0.593–0.616]** | **0.941 [0.937–0.945]** | **0.602 [0.595–0.609]** | **0.922 [0.919–0.926]** |

Performance was assessed by Spearman's rank correlation coefficient (rho) and root-mean-square error (RMSE). Data are reported as median [95% confidence intervals], calculated using 1,000 bootstraps.

this architecture to predict both time intervals simultaneously, often referred to as "multi-task learning", which aims to share knowledge amongst prediction tasks.

Next, we evaluated a broad spectrum of hyperparameter combinations for this network (S1 Table). This resulted in a multi-task LSTM architecture, consisting of three layers, including a dropout layer with a total of 56–104 neurons (S2 Table). During training, we used exponential learning-rate decay via the Adam optimization scheme [24]. The best validation results were achieved by use of an initial learning rate with a decay of 0.001 every 1,000 training steps, with a batch size of 1024, and a back-propagation through a time window of 30 minutes. This defines the amount of historic data the model uses, which in our case translates to six (first dataset) or 120 (second dataset) glucose data points, for the model to provide a prediction. The loss function during training was the mean average of the mean-squared error function of all predictions. The maximum amount of epochs was 50.000 with an early stopping criterion (based on 20% hold-out data) set to 250 epochs. We performed data preprocessing, model development, selection, and training using Python programming language (version 3.7.1) with the use of packages Numpy (version 1.17), Pandas (version 0.24), Keras (version 2.2.2), Scikit-learn (version 0.22.0) and Tensorflow (version 2.0.1, beta).

## Translation of the prediction models to the OhioT1DM Dataset

We used data from the OhioT1DM Dataset to explore whether our CGM-based prediction models would translate to individuals with type 1 diabetes. The OhioT1DM Dataset is freely available for scientific purposes and contains data of 6 individuals with type 1 diabetes who were all using insulin pump therapy and CGM [25]. The participants provided interstitial glucose values every five minutes for an eight-week period. First, in order to also include 30-minute prediction, we retrained our main CGM-based models on the main study population with identical hyperparameters and settings (S2 Table). Then, we evaluated the main CGM-based model on the test portion of the OhioT1DM Dataset (20%). Next, we aimed to optimize our main CGM-based model by training it on the train portion of the OhioT1DM Dataset.

Specifically, we trained the model using an Adam optimizer with a learning rate of $10^{-4}$, a batch size of 1024, a maximum of 10.000 epochs and an early stopping criterion (based on 20% of the training data) set to 100 epochs. Last, we evaluated this optimized model on the test portion using performance metrics and safety error grids, as described previously.

## Model evaluation and statistical analysis

Model evaluation was performed in the independent evaluation sets of individuals that were not used during model development (Fig 1, panel c). We employed several metrics to assess the performance of our models: root-mean-square error (RMSE), proportion of predicted values within 5% or 10% of actual glucose values, and Spearman's rank correlation coefficient (rho) (S2 File). Bootstrapping was performed to obtain 95% confidence intervals for each of these metrics [26]. In addition, we used error grids that are classically used for assessment of blood glucose monitor safety (i.e., surveillance error grid, Parkes error grid) to evaluate the safety of our glucose prediction models [27, 28]. Last, we performed several sensitivity analysis in our main study population by stratifying model performance for: (1) GMS (i.e., separate results for NGM and prediabetes); (2) day (06.00 to 24.00h) and night (24.00 to 06.00h); and (3) low or high glucose variability, defined as the 97.5th percentile of CGM-assessed SD in individuals with NGM (SD > 1.37 mmol/L) [14].

Normally distributed data are presented as mean ± SD, non-normally distributed data as median and interquartile range, and categorical data as n (%). Statistical analyses were performed using the Statistical Package for Social Sciences (version 25.0; IBM, Chicago, Illinois, USA) and the Python programming language (version 3.7.1).

## Results

### Main study population characteristics

In total, 896 individuals underwent CGM as part of The Maastricht Study's extensive phenotyping approach. We included participants with at least 48h of CGM data and either NGM, prediabetes, or type 2 diabetes. This resulted in the final study population of 851 individuals. Of this population, 540 participants (63.5%) simultaneously underwent CGM and accelerometry.

Table 2 shows the overall and type 2 diabetes-stratified characteristics of the two study populations (CGM-based as well as CGM- and accelerometry-based glucose prediction). The overall participant characteristics of both populations were generally comparable with regard to age, sex, BMI, glycaemic indices, blood pressure, and lipid profile, although the latter contained fewer participants with prediabetes or type 2 diabetes. Additionally, the participants with type 2 diabetes in the CGM- and accelerometry-based glucose prediction population were more often newly diagnosed with type 2 diabetes. Accordingly, these participants less often used glucose-lowering medication. Participant characteristics of the NGM and prediabetes subgroups are described in S3 Table.

### Overall performance of machine learning-based glucose prediction

We trained two machine learning models (i.e., CGM-based; CGM- and accelerometry-based) in predicting glucose levels at 15- and 60-minute intervals. Visually, both models appeared capable of accurately predicting the real glucose profiles, as illustrated by the representative examples in S1 and S2 Figs. Next, we assessed the performance of our models in our evaluation datasets with a variety of metrics, including an average error term (RMSE), the proportion of predictions within 5% or 10% deviation of the actual value, and correlation (rho). The

**Table 2. Participant characteristics of the CGM-based and CGM- and accelerometry-based glucose prediction study populations.**

| Characteristic | CGM-based glucose prediction | | CGM- and accelerometry-based glucose prediction | |
| --- | --- | --- | --- | --- |
| | Total (n = 851) | T2D (n = 197) | Total (n = 540) | T2D (n = 68) |
| Age, years | 59.9 ± 8.7 | 62.4 ± 7.8 | 59.1 ± 8.7 | 62.0 ± 6.9 |
| Women, n (%) | 418 (49.1) | 69 (35.0) | 276 (51.1) | 22 (32.4) |
| BMI, kg/m$^2$ | 27.2 ± 4.4 | 29.7 ± 4.7 | 26.5 ± 4.0 | 28.6 ± 4.1 |
| Newly diagnosed T2D, n (%) | 70 (8.2) | 70 (35.5) | 35 (6.5) | 35 (51.5) |
| Glucose metabolism status | | | | |
| NGM/PreD/T2D, n | 470/184/197 | - | 372/99/68 | - |
| NGM/PreD/T2D, % | 55.2/21.6/23.1 | - | 69.1/18.3/12.6 | - |
| Fasting plasma glucose, mmol/L | 5.4 [5.0–6.2] | 7.3 [6.5–8.4] | 5.3 [4.9–5.8] | 7.2 [6.3–8.4] |
| 2-h post-load glucose, mmol/L | 6.7 | 13.6 | 6.2 | 12.5 |
| | [5.2–9.1] | [11.7–16.2] | [5.0–7.7] | [11.3–16.6] |
| HbA$_{1c}$, % | 5.7 ± 0.8 | 6.7 ± 1.0 | 5.6 ± 0.6 | 6.4 ± 0.9 |
| HbA$_{1c}$, mmol/mol | 39.1 ± 8.3 | 49.2 ± 10.8 | 37.3 ± 6.2 | 46.9 ± 10.2 |
| Sensor glucose | | | | |
| Mean, mmol/L | 6.1 [5.7–6.7] | 7.5 [6.8–8.7] | 5.9 [5.6–6.4] | 7.3 [6.5–8.2] |
| SD, mmol/L | 0.84 | 1.51 | 0.79 | 1.46 |
| | [0.68–1.18] | [1.14–1.95] | [0.66–1.01] | [0.94–1.99] |
| SD > 1.37 mmol/L, n (%) | 142 (16.7) | 115 (58.4) | 50 (9.3) | 36 (52.9) |
| CV, % | 14.0 | 19.3 | 13.3 | 19.2 |
| | [11.6–17.6] | [15.9–24.0] | [11.2–16.8] | [14.5–24.1] |
| Diabetes medication use, n (%) | 109 (12.8) | 109 (55.6) | 27 (4.8) | 27 (39.7) |
| Insulin | 19 (2.2) | 19 (9.6) | 4 (0.7) | 4 (5.9) |
| Metformin | 104 (12.2) | 104 (53.1) | 27 (5.0) | 27 (39.7) |
| Sulfonylureas | 21 (2.5) | 21 (10.7) | 6 (1.1) | 6 (8.8) |
| Thiazolidinediones | 0 (0) | 0 (0) | 0 (0) | 0 (0) |
| GLP-1 analogues | 3 (0.4) | 3 (1.5) | 1 (0.2) | 1 (1.5) |
| DDP-4 inhibitors | 1 (0.1) | 1 (0.5) | 0 (0) | 0 (0) |
| SGLT-2 inhibitors | 1 (0.1) | 1 (0.5) | 0 (0) | 0 (0) |
| Office SBP, mmHg | 133.3 ± 18.0 | 139.4 ± 15.6 | 132.2 ± 17.9 | 137.7 ± 15.3 |
| Office DBP, mmHg | 75.2 ± 10.2 | 77.7 ± 10.5 | 74.7 ± 10.1 | 77.7 ± 9.6 |
| Antihypertensive medication use, n (%) | 305 (35.9) | 126 (64.3) | 162 (30.0) | 41 (60.3) |
| Total-to-HDL cholesterol ratio | 3.5 [2.8–4.3] | 3.6 [2.9–4.3] | 3.4 [2.8–4.3] | 3.7 [2.8–4.6] |
| Triglycerides, mmol/L | 1.3 [0.9–1.8] | 1.5 [1.0–2.1] | 1.2 [0.9–1.7] | 1.6 [1.0–2.3] |
| Lipid-modifying medication use, n (%) | 212 (24.9) | 115 (58.4) | 100 (18.5) | 39 (57.4) |
| Smoking status | | | | |
| Never/former/current, n | 327/415/106 | 67/104/26 | 214/253/70 | 19/36/13 |
| Never/former/current, % | 38.6/48.9/12.5 | 34.0/52.8/13.2 | 39.9/47.1/13.0 | 27.9/52.9/19.1 |

Data are reported as mean ± SD, median [interquartile range], or number (percentage [%]) as appropriate. CGM, continuous glucose monitoring; BMI, body mass index; T2D, type 2 diabetes; NGM, normal glucose metabolism; PreD, prediabetes; HbA1c, glycated haemoglobin A1c; SD, standard deviation; CV, coefficient of variation; GLP-1, glucagon-like peptide-1; DPP-4, dipeptidase-4; SGLT-2, sodium-glucose cotransporter 2; SBP, systolic blood pressure; DBP, diastolic blood pressure; HDL, high-density lipoprotein.

evaluation datasets comprise 20% of the original or stratified study populations and thus vary in sample size (n = 13–170).

Overall, our models demonstrated high prediction accuracy, supported by low RMSE values and high proportions of predicted glucose values within 5% and 10% deviation (Table 3).

**Table 3. Overall performance in the main study population of CGM-based and CGM- and accelerometry-based machine learning models trained in predicting glucose values at time intervals of 15 and 60 minutes.**

| | | CGM-based glucose prediction | | CGM- and accelerometry-based glucose prediction | |
|---|---|---|---|---|---|
| | | Total (n = 170) | T2D (n = 43) | Total (n = 109) | T2D (n = 13) |
| **15 minutes** | RMSE, mmol/L | 0.188 [0.186–0.191] | 0.288 [0.281–0.306] | 0.184 [0.177–0.189] | 0.271 [0.260–0.282] |
| | < 5%, % | 92.98 [92.87–93.05] | 92.02 [91.83–92.25] | 93.06 [93.03–93.09] | 92.04 [91.99–92.11] |
| | < 10%, % | 99.17 [99.13–99.23] | 98.88 [98.82–98.94] | 99.25 [99.21–99.28] | 98.90 [98.83–98.97] |
| | Rho | 0.961 [0.959–0.962] | 0.987 [0.985–0.989] | 0.968 [0.964–0.970] | 0.990 [0.988–0.993] |
| **60 minutes** | RMSE, mmol/L | 0.589 [0.582–0.592] | 0.701 [0.692–0.711] | 0.582 [0.579–0.586] | 0.700 [0.693–0.708] |
| | < 5%, % | 70.22 [70.09–70.41] | 66.23 [66.13–66.33] | 70.11 [70.05–70.17] | 66.17 [66.09–66.22] |
| | < 10%, % | 87.39 [87.24–87.53] | 85.82 [85.70–85.93] | 87.44 [87.38–87.50] | 86.11 [86.01–86.20] |
| | Rho | 0.721 [0.719–0.722] | 0.781 [0.779–0.782] | 0.725 [0.721–0.729] | 0.790 [0.782–0.799] |

Data are reported as mean [95% confidence interval]. CGM, continuous glucose monitoring; T2D, type 2 diabetes; RMSE, root-mean-square error; < 5%, percentage of predicted values within 5% of actual glucose values; < 10%, percentage of predicted values within 10% of actual glucose values; rho, Spearman's rank correlation coefficient.

Model performance in the type 2 diabetes subgroup was generally lower compared to the overall group, except for correlation coefficients, which were often higher in individuals with type 2 diabetes. This phenomenon can be largely attributed to the lower correlation coefficients of individuals with NGM and prediabetes (S4 Table), which is caused by range restriction (i.e., smaller glucose ranges attenuate the correlation coefficients) [29]. Consequently, the correlation coefficients are valid for the comparison of CGM-based glucose prediction to CGM- and accelerometry-based glucose prediction, but not for comparison of the overall study population to the type 2 diabetes subgroup. In addition, we observed short-to-moderate time lags for the 15- and 60-minute predictions (S5 Table).

In general, incorporation of accelerometry data in the models only slightly improved performance metrics at both prediction intervals (Table 3). S4 Table shows the model performance in NGM and prediabetes subgroups. Glucose prediction was most precise in individuals with NGM. Of note, the ML-based models substantially outperformed a naive approach that used $t_0$ as predicted glucose value (S6 Table, S3 and S4 Figs).

## Safety evaluation with clinical error grids

We assessed the safety of our machine learning-based glucose prediction using two clinical error grids (i.e., surveillance and Parkes error grids). Fig 2 depicts the safety results for individuals with type 2 diabetes according to the surveillance error grid. At the 15-minute interval, almost all predictions (>99.9%) were clinically safe (i.e., a risk score between 0 and 1.0) (Fig 2, panels A and B). At the extended prediction window of 60 minutes, clinical safety was slightly lower (98.4–99.2%) (Fig 2, panels C and D). Parkes error grid assessment yielded similar results (S5 Fig). Of note, less accurate predictions were more often in the vertical B-D zones than in the horizontal B-E zones (e.g., S4 Fig, panel C: 11.80% versus 4.24%), which suggests a model tendency to underestimate rather than overestimate actual glucose values, the latter of which being more dangerous.

## Additional analyses

To further obtain insights into our model predictions, we assessed performance metrics stratified by day and night (S7 Table). Fifteen-minute predictions did not materially differ between day and night. By contrast, accuracy of 60-minute predictions was lower during the day than

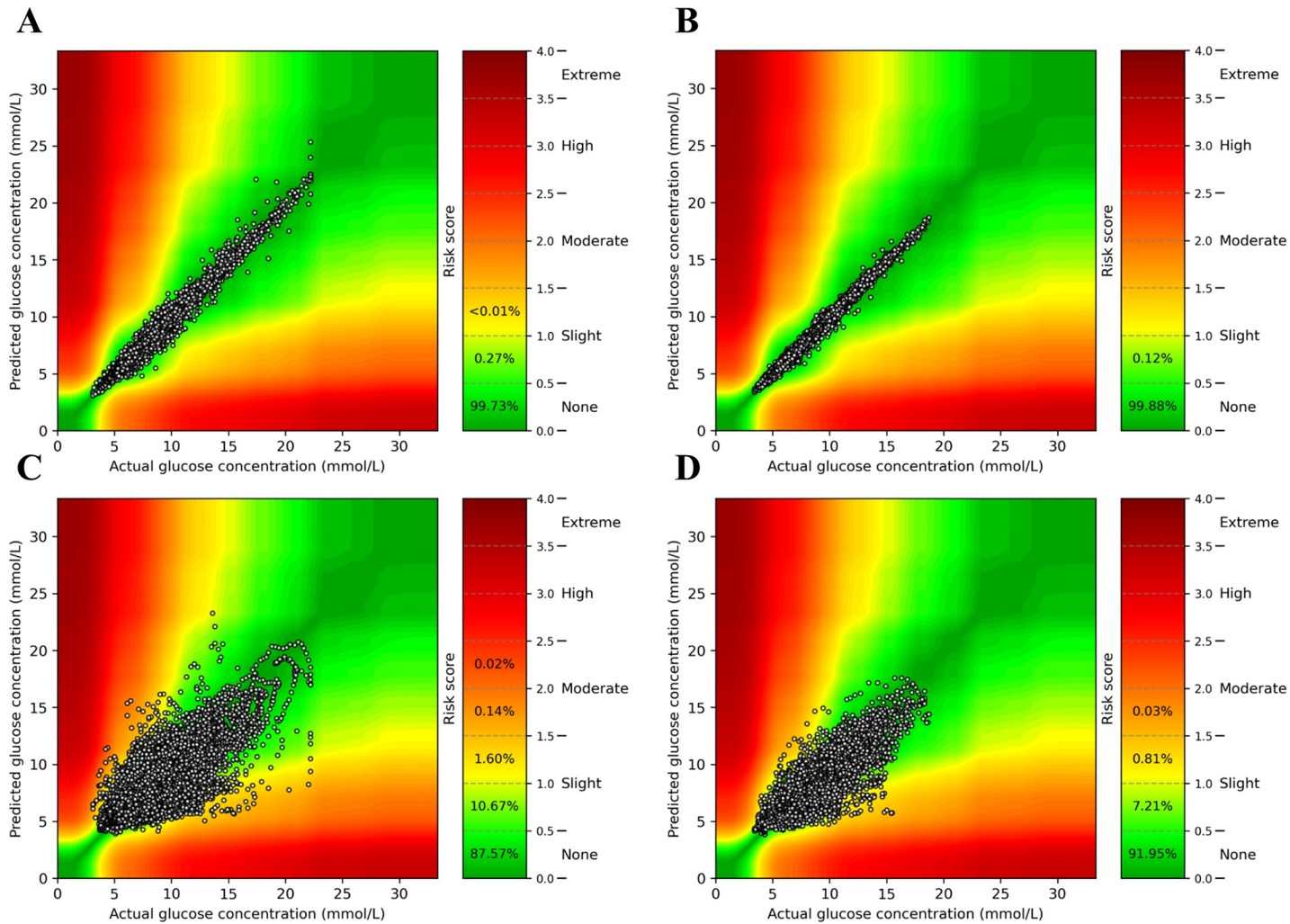

**Fig 2. Surveillance error grid evaluation of glucose prediction safety at time intervals of 15 and 60 minutes in the main study population.** Assessment of CGM-based glucose prediction safety in individuals with type 2 diabetes (n = 43) at 15 minutes (panel A) and 60 minutes (panel C). Assessment of CGM- and accelerometry-based glucose prediction safety in individuals with type 2 diabetes (n = 13) at 15 minutes (panel B) and 60 minutes (panel D). The risk score values translate to the following degrees of risk: 0–0.5, none; 0.5–1.0, slight (lower); 1.0–1.5, slight (higher); 1.5–2.0, moderate (lower); 2.0–2.5, moderate (higher); 2.5–3.0, great (lower); 3.0–3.5, great (higher); > 3.5 extreme [27].

at night. In addition, we stratified the results by high or low glucose variability (i.e., SD cut-off of 1.37 mmol/L) (S8 Table). Model performance was slightly lower at higher glucose variability, at both time intervals of 15 and 60 minutes.

## Translation of the prediction models to the OhioT1DM Dataset

The prediction accuracy of the CGM-based model that was developed with our main study population was moderate in individuals with type 1 diabetes (RMSEs at 15, 30, and 60 min: 0.689 [0.685–0.693], 1.189 [1.183–1.195], and 1.918 [1.910–1.926] mmol/L), but substantially improved after being trained on data from each individual with type 1 diabetes (RMSEs at 15, 30, and 60 min: 0.426 [0.422–0.430], 1.046 [1.039–1.052], and 1.733 [1.725–1.741] mmol/L; S9 Table). Accordingly, clinical safety was substantial as shown by the high percentages of clinically safe predictions (15-minute: >99%, 30-minute: >97%, and 60-minute: >91%; Fig 3).

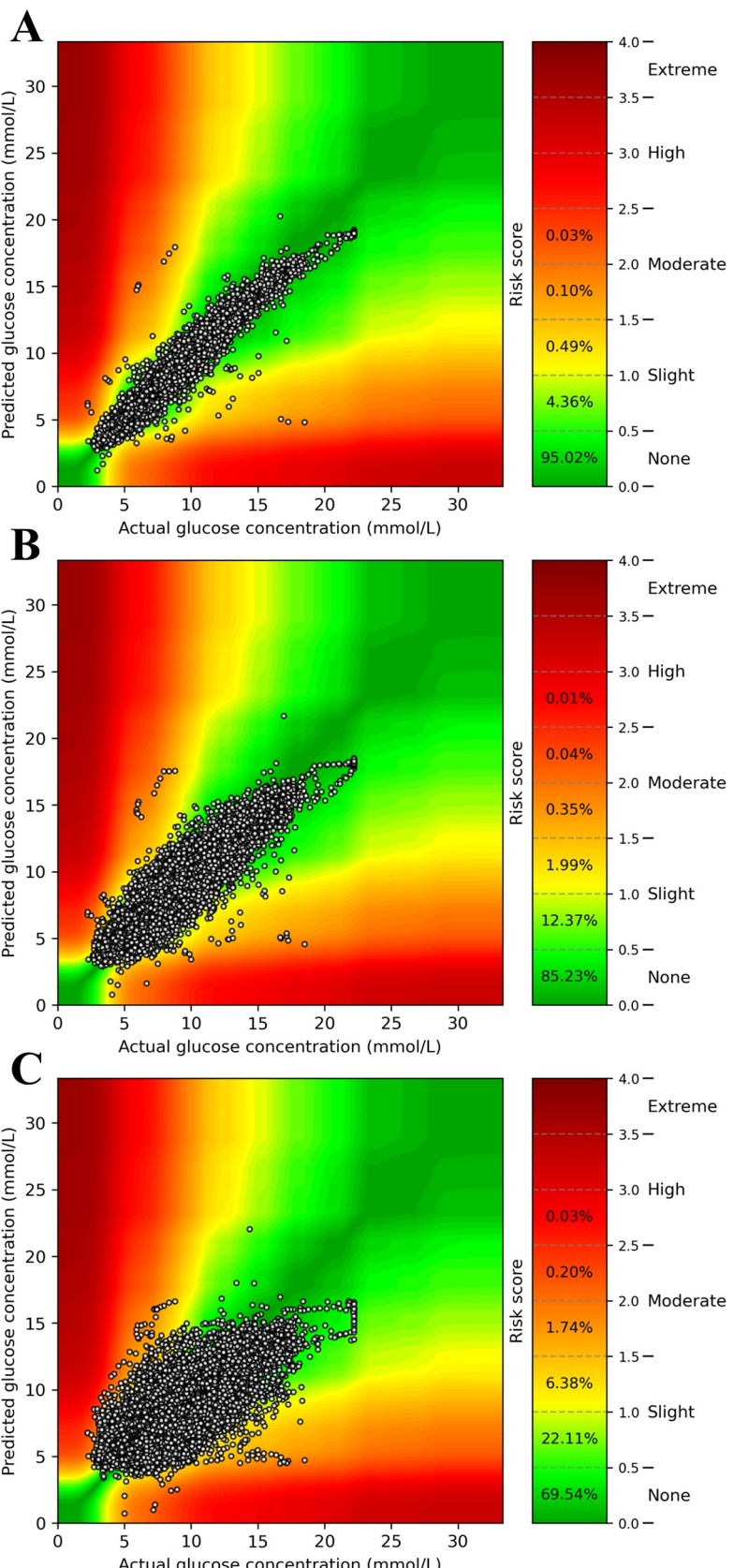

**Fig 3. Surveillance error grid evaluation of glucose prediction safety at time intervals of 15, 30, and 60 minutes in individuals with type 1 diabetes.** Assessment of CGM-based glucose prediction safety in individuals with type 1 diabetes (n = 6) at 15 (panel A), 30 (panel B), and 60 minutes (panel C). The risk score values translate to the following degrees of risk: 0–0.5, none; 0.5–1.0, slight (lower); 1.0–1.5, slight (higher); 1.5–2.0, moderate (lower); 2.0–2.5, moderate (higher); 2.5–3.0, great (lower); 3.0–3.5, great (higher); > 3.5 extreme [27].

## Discussion

In this study with 851 individuals and almost 1.4 million glucose measurements, we investigated whether glucose values can be accurately predicted by using machine learning-based models that utilise recently measured CGM and physical activity data with the prospect of improving closed-loop insulin delivery systems. Our study has several important findings and unique characteristics. First, the machine learning-based models are capable of accurately predicting the actual glucose profiles at 15 minutes, as reflected by several objective performance metrics (e.g., RMSE, rho; Table 2) and visual illustrations (S1 and S2 Figs). Despite prediction accuracy being moderately lower at 60 minutes, more than 98% of the predicted values remained sufficiently accurate to be deemed clinically safe based on surveillance error grids (Fig 2). Second, glucose prediction only improved slightly when accelerometer-assessed physical activity data was incorporated in the models. Third, translation of our CGM-based glucose prediction models to individuals with type 1 diabetes yielded encouraging results (i.e., ample prediction accuracy and clinical safety).

Although most research has thus far focused on type 1 diabetes [12], several efforts have been made to use machine learning for glucose prediction in individuals with type 2 diabetes [30–34]. Most of these studies assessed technical aspects of glucose prediction in relatively small (n = 1 to 50) or even virtual, *in silico* populations. Such studies provide valuable comparisons of models, but show suboptimal and highly variable performance in predicting glucose values. To our knowledge, this is the first study to report this level of performance in a large, population-based sample of individuals with NGM, prediabetes, or type 2 diabetes. Our CGM-based models were able to accurately predict glucose values at 15 (RMSEs, overall/type 2 diabetes: 0.19/0.29 mmol/L) and 60 minutes (RMSEs, overall/type 2 diabetes: 0.59/0.70 mmol/L). These results surpass previously reported RMSE values for a sample of 50 individuals with type 2 diabetes, which were 0.65 and 1.50 mmol/L for 15- and 60-minute CGM-based glucose prediction, respectively [34]. We expect this difference to, in part, stem from our much larger sample size. To our knowledge, our exploratory translation to individuals with type 1 diabetes (S9 Table) showed that our models perform equally well as recent publications in the field [12, 35–38]. For example, the best performing model of the Blood Glucose Level Prediction Challenge 2018, which was also based on a LSTM architecture as well as was trained on and evaluated in the OhioT1DM Dataset, reported 30-minute and 60-minute RMSEs of 1.05 and 1.74 mmol/L [35]. Additionally, Kriventsov et al. recently described large-scale application of glucose prediction in a smartphone app (Diabits) and reported a comparable RMSE at 30 minutes (1.04 mmol/L) [36]. We anticipate that further technical development of our prediction models, while using a larger sample of individuals with type 1 diabetes, will advance performance even more.

We integrated physical activity, which we assessed via accelerometry, into our glucose prediction model, because of its short- and long-term effects on daily glucose patterns. Whereas an acute bout of physical activity can either decrease or increase serum glucose levels, prolonged exercise improves insulin sensitivity, and thus insulin-stimulated glucose uptake [39]. While it should be noted that CGM- and accelerometry-based glucose prediction yielded larger improvements relative to CGM-based glucose prediction for the 60-minute interval, most notably during the day (S7 Table) and in individuals with higher glucose variability (S9 Table), incorporation of physical activity generally only marginally improved glucose

prediction. This can be explained by the observation that the models based on CGM data only already performed very well, which limits the ability to achieve additional improvements [40]. Also, the effect of physical activity on serum glucose levels is relatively small in people with beta-cell function that is either normal or only mildly deficient. Given the absence of pancreatic glucor-egulation in individuals with type 1 diabetes, it is conceivable that incorporation of accelerometry data leads to more substantially improved model performance in this patient group [40], which, at present, we were not able to further explore. In addition, a time interval of 15 or 60 minutes could be too short to incorporate long-term physical activity effects into the prediction model.

The closed-loop insulin delivery system has been shown to improve glycaemic control in individuals with type 1 or type 2 diabetes [8, 9, 41]. Nevertheless, several aspects of the artificial pancreas require further enhancement [6, 10]. Our results demonstrate that machine learning-based glucose prediction has the promise of being a valid and safe strategy to both overcome ~10-minute sensor delay and bridge prolonged periods of sensor malfunction. Not only are more than 99% of the predicted glucose values in clinically safe zones (i.e., Parkes error grid zone A and B), the model also tended to slightly underestimate rather than overestimate the actual glucose values. In case the prediction model were to be implemented, this would further reduce the risk of iatrogenic hypoglycaemia. Nevertheless, future research is needed to assess whether incorporation of these prediction models in a closed-loop insulin delivery system safely improves glycaemic control.

This proof-of-principle study has several strengths and limitations. Strengths are 1) the largest well-characterized, population-based study sample thus far, which ensured sufficient statistical power; 2) the unique large-scale combination of CGM and continuous accelerometry, which enabled us to study to what extent incorporation of data on physical activity would improve prediction in this population; 3) the gold-standard assessment of GMS, which allowed for the comparison of performance in NGM, prediabetes and type 2 diabetes; 4) the broad and solid evaluation of various statistical and machine learning architectures for this prediction task; and 5) result robustness, as reflected by the consistency of several statistical and clinical performance metrics.

Our research had certain limitations. First, the main study population comprised individuals with NGM, prediabetes, or type 2 diabetes, who are generally not the target population for closed-loop insulin delivery systems. We, therefore, exploratively investigated whether our prediction models would translate to individuals with type 1 diabetes using the OhioT1DM Dataset, which yielded encouraging results. Nevertheless, we underscore the importance of extensive evaluation of the models in a larger sample of individuals with type 1 diabetes, insulin-treated type 2 diabetes, or both. Second, we were unable to factor in other important elements pertaining to glycaemic control (e.g., diet or medication use) [6]. In automated, self-regulatory closed-loop systems, utilization of these kinds of data requires manual input, which is less convenient and reliable than CGM. In addition, since glucose prediction was only slightly improved by incorporating physical activity, we expect relatively little gain from including such factors into our models, at least in individuals with type 2 diabetes. However, given the results of several small studies that have incorporated diet and medication use [12], we acknowledge that this may not hold true for individuals with type 1 diabetes. In this regard, large-scale studies are required to reach more definitive conclusions. If diet, medication use, or other factors were to be incorporated, it is necessary to evaluate whether LSTM remains the best-performing machine learning architecture.

## Conclusion

In this study, we show that our machine learning-based models are able to accurately and safely predict glucose values for up to 60 minutes in individuals with, NGM, prediabetes, or

type 2 diabetes. In addition, translation of our prediction models to individuals with type 1 diabetes showed encouraging results. We observed particularly high precision at a 15-minute prediction window, which is a clinically relevant timespan to align interstitially measured glucose values by continuous glucose measurement systems with actual plasma glucose values. As such, the prediction model can be used to improve closed-loop insulin delivery systems by overcoming sensor delay. In addition, longer prediction intervals may be used to safely bridge periods of sensor malfunction. Last, our current findings question the use of accelerometry to substantially improve prediction. Future research should validate our findings by replicating the results in a larger sample of individuals with type 1 diabetes and studying the effects of implementing the prediction model in a closed-loop insulin delivery system.

## Supporting information

**S1 Fig. Illustrative examples of continuous glucose monitoring-based machine learning model predictions compared to actual values.**
(DOCX)

**S2 Fig. Illustrative examples of continuous glucose monitoring- and accelerometry-based machine learning model predictions compared to actual values.**
(DOCX)

**S3 Fig. Surveillance error grid evaluation of glucose prediction safety at time intervals of 15 and 60 minutes using glucose value t0 as predictor.**
(DOCX)

**S4 Fig. Performance characteristics of a prediction model using t0 as predictor across time horizons between 0 and 120 minutes.**
(DOCX)

**S5 Fig. Parkes error grid evaluation of glucose prediction safety at time intervals of 15 and 60 minutes.**
(DOCX)

**S1 Table. Hyperparameter combinations evaluated in current experiments.**
(DOCX)

**S2 Table. Final set of hyperparameters for each of the machine learning models.**
(DOCX)

**S3 Table. Extended baseline characteristics.**
(DOCX)

**S4 Table. Extended analysis of model performance in normal glucose metabolism and pre-diabetes subgroups.**
(DOCX)

**S5 Table. Extended analysis on time lag between predicted and actual glucose values.**
(DOCX)

**S6 Table. Extended analysis of model performance with t0 glucose value as predictor.**
(DOCX)

**S7 Table. Model performance stratified by day and night.**
(DOCX)

**S8 Table. Model performance stratified by low versus high glucose variability.**
(DOCX)

**S9 Table. Extended analysis of model performance in the Ohio T1DM Dataset.**
(DOCX)

**S1 File. Background information on machine learning models reviewed in current study.**
(DOCX)

**S2 File. Background information on metrics used in the current study.**
(DOCX)

## Acknowledgments

**Prior presentation**

An abstract of this study was submitted to the Annual Meeting of the European Association for the Study of Diabetes (Vienna, Austria, 21–25 September 2020). The conference abstract has been published by EMJ Diabetes.

## Author Contributions

**Conceptualization:** William P. T. M. van Doorn, Ronald M. A. Henry, Otto Bekers, Coen D. A. Stehouwer, Steven J. R. Meex, Martijn C. G. J. Brouwers.

**Data curation:** Nicolaas C. Schaper, Bastiaan E. de Galan.

**Formal analysis:** William P. T. M. van Doorn, Yuri D. Foreman, Annemarie Koster, Carla J. H. van der Kallen, Miranda T. Schram, Steven J. R. Meex, Martijn C. G. J. Brouwers.

**Investigation:** William P. T. M. van Doorn, Steven J. R. Meex.

**Methodology:** William P. T. M. van Doorn, Yuri D. Foreman, Annemarie Koster, Ronald M. A. Henry, Coen D. A. Stehouwer, Steven J. R. Meex, Martijn C. G. J. Brouwers.

**Project administration:** Yuri D. Foreman.

**Resources:** Coen D. A. Stehouwer, Steven J. R. Meex, Martijn C. G. J. Brouwers.

**Software:** Martijn C. G. J. Brouwers.

**Supervision:** Nicolaas C. Schaper, Hans H. C. M. Savelberg, Anke Wesselius, Miranda T. Schram, Ronald M. A. Henry, Pieter C. Dagnelie, Bastiaan E. de Galan, Otto Bekers, Coen D. A. Stehouwer, Steven J. R. Meex, Martijn C. G. J. Brouwers.

**Writing – original draft:** William P. T. M. van Doorn, Yuri D. Foreman, Otto Bekers, Coen D. A. Stehouwer, Steven J. R. Meex, Martijn C. G. J. Brouwers.

**Writing – review & editing:** William P. T. M. van Doorn, Yuri D. Foreman, Nicolaas C. Schaper, Hans H. C. M. Savelberg, Annemarie Koster, Carla J. H. van der Kallen, Anke Wesselius, Miranda T. Schram, Ronald M. A. Henry, Pieter C. Dagnelie, Bastiaan E. de Galan, Otto Bekers, Coen D. A. Stehouwer, Steven J. R. Meex, Martijn C. G. J. Brouwers.

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
