## [Decision Letter · Decision Letter 0]

17 Dec 2020

PONE-D-20-30681

Machine learning-based glucose prediction with use of continuous glucose and physical activity monitoring data: The Maastricht Study

PLOS ONE

Dear Dr. Brouwers,

Thank you for submitting your manuscript to PLOS ONE. After careful consideration, we feel that it has merit but does not fully meet PLOS ONE’s publication criteria as it currently stands. Therefore, we invite you to submit a revised version of the manuscript that addresses the points raised during the review process.

We look forward to receiving your revised manuscript.

Kind regards,

Chi-Hua Chen, Ph.D.

Academic Editor

PLOS ONE

Journal Requirements:

2. Thank you for stating the following in the Financial Disclosure * (delete as necessary) section:

"The Maastricht Study was supported by the European Regional Development Fund via OP-Zuid, the Province of Limburg, the Dutch Ministry of Economic Affairs (grant 31O.041), Stichting De Weijerhorst (Maastricht, the Netherlands), the Pearl String Initiative Diabetes (Amsterdam, the Netherlands), School for Cardiovascular Diseases (CARIM, Maastricht, the Netherlands), School for Public Health and Primary Care (CAPHRI, Maastricht, the Netherlands), School for Nutrition and Translational Research in Metabolism (NUTRIM, Maastricht, the Netherlands), Stichting Annadal (Maastricht, the Netherlands), Health Foundation Limburg (Maastricht, the Netherlands), and by unrestricted grants from Janssen-Cilag B.V. (Tilburg, the Netherlands), Novo Nordisk Farma B.V. (Alphen aan den Rijn, the Netherlands), Sanofi-Aventis Netherlands B.V. (Gouda, the Netherlands), and Medtronic (Tolochenaz, Switzerland). The funders had no role in study design, data collection and analysis, decision to publish, or preparation of the manuscript."

We note that you received funding from a commercial source: Janssen-Cilag B.V., Novo Nordisk Farma B.V., Sanofi-Aventis Netherlands B.V., Medtronic.

Reviewers' comments:

Reviewer's Responses to Questions

**Comments to the Author**

1. Is the manuscript technically sound, and do the data support the conclusions?

Reviewer #1: Yes

Reviewer #2: Partly

Reviewer #3: No

2. Has the statistical analysis been performed appropriately and rigorously? 

Reviewer #1: Yes

Reviewer #2: No

Reviewer #3: No

3. Have the authors made all data underlying the findings in their manuscript fully available?

Reviewer #1: Yes

Reviewer #2: No

Reviewer #3: No

4. Is the manuscript presented in an intelligible fashion and written in standard English?

Reviewer #1: Yes

Reviewer #2: Yes

Reviewer #3: Yes

5. Review Comments to the Author

Reviewer #1: One issue of people with T2D using CGM is that, usually they do not need a CGM daily to monitor their glucose level all the time because they do not need to inject insulin like people with T1D. Please address this point to clarify the motivation and contribution of this work.

The references cited in this paper is not state-of-the-art. Many important related references using CGM, wearables in diabetes management using machine learning, are missing, such as

Data-driven modeling and prediction of blood glucose dynamics: Machine learning applications in type 1 diabetes

Convolutional recurrent neural networks for glucose prediction

Prediction of hypoglycemia during aerobic exercise in adults with type 1 diabetes

Normally people investigate the prediction of next 15, 30 and 60 mins. Why only 15 and 60 mins results were discussed in this paper

How to you deal with meal and insulin data in the prediction model? If they are not included, it seems the prediction can merely follow the trend of real glucose value to achieve an acceptable accuracy.

Not clear about the dataset. It says ‘From September 19, 2016 until September 13, 2018, participants were invited to undergo CGM.’ So how many dates of CGM data does the dataset have?

People cannot tell details in Figure S1, S2. Could you please zoom in so readers can see the difference? In addition, it is better to compare the results of different algorithms in figures.

How the extra accelerometer data contribute the accuracy of glucose prediction, co

mparing to the accuracy of sole CGM-based glucose prediction? Happy to see a concrete discussion to address this.

Besides RMSE, can you please calculate the time lag between the real and predicted glucose curve, in terms of different algorithms used in the paper? Because it is an important feature to measure the performance.

The results of RMSE (at 15 (RMSE:0.19mmol/L; rho:0.96) and 60 minutes (RMSE:0.59mmol/L, rho:0.72).), are too good to be true, from my point of view. For example, even give meal and insulin, exercise info, the RMSE of 60 mins prediction for T1D is larger than 30 mg/dL. For T2D the results will be better, but 0.59mmol/L is still very small. Can you compare your results to other existing algorithms, and convince readers that this good results is in feasible.

Reviewer #2: The study proposes a straight forward strategy of predicting blood glucose levels using ML models. The models are trained with a large dataset of 851 patients. The dataset contains data from T2Ds, prediabetics and normal individuals. The forecasting is done for a PH of 15 and 60 minutes. The results show almost perfect prediction, this is due to methodological errors.

The authors claim to have split training, cross-validation and test data randomly. This could prove to be a wrong strategy in time-series forecasting as there is a chance of the model getting trained on the future data.

The authors trained multiple models for prediction purpose. It is seen in the performance comparison table that classical RNN performs best for 15 min PH and LSTM performs best for 60 min PH. The manuscript, however, only contains details about the LSTM model.

Since, the proposed study does exactly the same what various other works have been doing for BG prediction during the past decade, no attempt at performance comparison with prior work has been made.

Performance improvement depicted in the CGM+PA dataset is not significant and hence provides no motive for designers to prefer one over the other.

Since it is understood that the glucose variability in NGM is low, and the number of individuals with NGM in both datasets are the largest, the underlying trends being identified by the ML model are overwhelmed by such data. It explains why the ML model are predicting almost perfectly.

Reviewer #3: This article presents the work on the application of different machine learning techniques for glucose prediction from CGM and physical activity bracelets.

This is a study with a very large number of patients but in my opinion the article is not interesting for the journal for several reasons.

Firstly, the data are not available to the research community, which makes it difficult to check whether the techniques presented can be overcome by the countless number of papers in the area.

Secondly, no new techniques are proposed, there are many studies in the area and the techniques of machine learning have been studied in depth, the authors can see for example all the articles in results were reported in different previous publications recently and some years ago. You can see, for instance the works presented at the last two workshops on Blood Glucose Level Prediction (BGLP) Challenge

http://ceur-ws.org/Vol-2675/

You can also find several journal papers

Hidalgo, J. I., Colmenar, J. M., Kronberger, G., Winkler, S. M., Garnica, O., & Lanchares, J. (2017). Data based prediction of blood glucose concentrations using evolutionary methods. Journal of medical systems, 41(9), 142.

Woldaregay, A. Z., Årsand, E., Walderhaug, S., Albers, D., Mamykina, L., Botsis, T., & Hartvigsen, G. (2019). Data-driven modeling and prediction of blood glucose dynamics: Machine learning applications in type 1 diabetes. Artificial intelligence in medicine, 98, 109-134.

Velasco, J. M., Garnica, O., Lanchares, J., Botella, M., & Hidalgo, J. I. (2018). Combining data augmentation, EDAs and grammatical evolution for blood glucose forecasting. Memetic Computing, 10(3), 267-277.

De Falco, I., Della Cioppa, A., Giugliano, A., Marcelli, A., Koutny, T., Krcma, M., ... & Tarantino, E. (2019). A genetic programming-based regression for extrapolating a blood glucose-dynamics model from interstitial glucose measurements and their first derivatives. Applied Soft Computing, 77, 316-328.

Contreras, I., & Vehi, J. (2018). Artificial intelligence for diabetes management and decision support: literature review. Journal of medical Internet research, 20(5),

And even more from the last two years on NNs and DL approaches

Moreover a prediction horizon of 15 minutes is so short that any Naive approach could reach a 95% of safe predictions, I recommend the authors the exercise of predicting the glucose value for 15 minutes as the value a t=0.

Experimental results are not useful as they are presented in the paper. All the techniques are summarized in just one table and no discrimination among them is done. The main conclusion of the paper is so general that is obvious. Is something like the affirmation " Medicine is good" or something similar.

Last but no least, the study affirm that, although it was made with T2 diabetes patients, it could be extrapolated to other T1 patients. I am sure that conclusions for T2 can not directly extrapolated to other type of patients. It has been shown in the past that in-silico results are not extensible to T1 real patients nor to T2 and vice versa. Glucose Variability of one T1 or T2 patients are different, T1 can produce little amounts of insulin or not, T2 insulin resistance could be heavier for one patient than for other....

So in this conditions the study is of little interest for the journal, In my humble opinion, I think that the data set has a great potential and that the research team is capable of prepare and in depth analysis of machine learning technique, I would recommend to separate and configure ML techniques for the different types of patients, and of course when presenting the results separate by ML techniques and data sets.

6. PLOS authors have the option to publish the peer review history of their article (what does this mean?). If published, this will include your full peer review and any attached files.

Reviewer #1: No

Reviewer #2: No

Reviewer #3: No

---

## [Author Response · Author response to Decision Letter 0]

8 Feb 2021

We would like to thank the reviewers for their positive feedback on our study and for the time spent on our manuscript. We believe that their comments have given us the opportunity to substantially improve our manuscript. Notably, we have now included a first step of model validation in individuals with type 1 diabetes (OhioT1DM Dataset). As the reviewers can appreciate, our prediction models translate quite well to individuals with type 1 diabetes and are competitive with current studies in type 1 diabetes (S10 Table and Figure 3). Please find below our point-by-point rebuttal.

Reviewer #1

1. One issue of people with T2D using CGM is that, usually they do not need a CGM daily to monitor their glucose level all the time because they do not need to inject insulin like people with T1D. Please address this point to clarify the motivation and contribution of this work.

We certainly agree with the reviewer on this point. At present, most of the individuals with type 2 diabetes do not have an indication to wear CGM for a long time period. Furthermore, the number of individuals with type 2 diabetes who have an indication for a closed-loop insulin delivery system is even lower, although the use of such systems in type 2 diabetes has been investigated(1) and may become more frequent in the future. 

Hence, we acknowledge that individuals with type 1 diabetes are, at present, the main target population for closed-loop insulin delivery systems, and had already detailed this in the limitations part of our Discussion (Page 18, Lines 395-397). We have now added the point that individuals with type 1 diabetes are the main population eligible for closed-loop insulin delivery to our Introduction (Page 6, Lines 103-105).

2. The references cited in this paper is not state-of-the-art. Many important related references using CGM, wearables in diabetes management using machine learning, are missing, such as Data-driven modeling and prediction of blood glucose dynamics: Machine learning applications in type 1 diabetes; Convolutional recurrent neural networks for glucose prediction; Prediction of hypoglycemia during aerobic exercise in adults with type 1 diabetes

Since we have now expanded our work to include translation of our prediction models to individuals with type 1 diabetes, and have updated our references accordingly, we would like to thank the reviewer for the literature suggestions.

3. Normally people investigate the prediction of next 15, 30 and 60 mins. Why only 15 and 60 mins results were discussed in this paper

Indeed, several previous studies have included 30 minutes as a prediction interval in addition to the 15 and 60 minutes that we used for our main results. Our reasoning for this was based on clinical applicability: 15 minutes to reflect overcoming sensor delay (i.e., the inherent ~10-minute discrepancy between interstitially measured and actual plasma glucose values) and 60 minutes to reflect bridging a relatively long period of sensor malfunction. We have outlined this in our Introduction (Pages 5-6, Lines 81-93). Moreover, if prediction accuracy and clinical safety are very high at 60 minutes, there is little reason to also investigate 30 or 45 minutes.

However, as advised by the reviewer, we have included the 30-minute interval for the prediction model validation in individuals with type 1 diabetes (S10 Table) in order to allow comparison with the current literature. This turned out to be more justified for this population, since clinical prediction safety was not as high at 60 minutes as compared to type 2 diabetes. 

4. How to you deal with meal and insulin data in the prediction model? If they are not included, it seems the prediction can merely follow the trend of real glucose value to achieve an acceptable accuracy.

Unfortunately, we were unable to incorporate meal and therapy data in our prediction models, since they were not available for the seven-day recording period. We have discussed this in the Limitations section (Pages 18-19, Lines 421-431). We agree that incorporation of these data would seem logical from a physiological viewpoint. However, in automated, self-regulatory closed-loop systems, it would require manual input. Also, prediction in individuals with type 2 diabetes was accurate and safe to such an extent that addition of meal and therapy data is expected to lead to only a slight improvement of the prediction models. Indeed, the 15- and 60-minute prediction models are based on only previous glucose values (in case of the CGM-based approach), but we do not regard this as problematic, since the accuracy and clinical safety are nevertheless high. Still, we acknowledge that this may be different for individuals with type 1 diabetes (Page 18-19, Lines 417-421).

5. Not clear about the dataset. It says ‘From September 19, 2016 until September 13, 2018, participants were invited to undergo CGM.’ So how many dates of CGM data does the dataset have?

We apologize for the misunderstanding. The sentence referred to the total inclusion period of participants. All participants underwent seven-day CGM. We have rewritten this part of the study inclusion process and moved it to the Study population and design (Page 7, Lines 117-124). 

6. People cannot tell details in Figure S1, S2. Could you please zoom in so readers can see the difference? In addition, it is better to compare the results of different algorithms in figures.

We adjusted S1 and S2 Figure to include a certain region that is zoomed in, so the actual and predicted glucose profiles can be examined. To ensure a fair comparison, we also reduced the line width of both profiles by a small margin.

7. How the extra accelerometer data contribute the accuracy of glucose prediction, comparing to the accuracy of sole CGM-based glucose prediction? Happy to see a concrete discussion to address this.

As already discussed on Pages 18-19 (Lines 378-393), we propose the following explanations for the very modest improvement in the prediction model after incorporating the accelerometer data: 1) the CGM-only models perform very well, and as such, substantial further improvement is very difficult to achieve; 2) the contribution of accelerometer data may physiologically be greater in type 1 diabetes (which, unfortunately, we were not able to investigate at present); and 3) the time intervals used may be too short to for the model to incorporate sustained physical activity effects into the prediction.

8. Besides RMSE, can you please calculate the time lag between the real and predicted glucose curve, in terms of different algorithms used in the paper? Because it is an important feature to measure the performance.

As suggested by the reviewer, we calculated the time lag between the real and predicted glucose value. We calculated the prediction time lag by measuring the time-shift that results in the highest cross correlation coefficient between them, according to the formula(2, 3):

τ_delay= 〖arg max┬k〗⁡〖 ((y_k ) ˇ(k│k-PH)*y(k))〗

These results have now been added to the supplemental materials (S6 Table) and we have updated our S2 Supporting information accordingly.

9. The results of RMSE (at 15 (RMSE:0.19mmol/L; rho:0.96) and 60 minutes (RMSE:0.59mmol/L, rho:0.72).), are too good to be true, from my point of view. For example, even give meal and insulin, exercise info, the RMSE of 60 mins prediction for T1D is larger than 30 mg/dL. For T2D the results will be better, but 0.59mmol/L is still very small. Can you compare your results to other existing algorithms, and convince readers that this good results is in feasible.

It should be acknowledged that our study was based on individuals with normal glucose metabolism (NGM), prediabetes, and type 2 diabetes, not type 1 diabetes. Therefore, we primarily compared our results to studies with a comparable study population (i.e., individuals with type 2 diabetes). The current literature in type 2 diabetes is limited, which may explain why our results seem so good. Nevertheless, when comparing our results to the best algorithm in type 2 diabetes published to date (with at least a sample size > 10 participants)(4), our prediction results are indeed better. We expect this to be mainly due to a large sample size difference (i.e., a larger sample yields better and more reliable prediction), which we have now added to the Discussion section (Page 18, Lines 365-366). Moreover, as our approach ensured that we evaluated our models in individuals completely retained from model development, we prevented our models from recognizing data patterns on which they were trained.

As we have now added an exploratory validation in individuals with type 1 diabetes, we have added a comparison of these result to the current literature on individuals with type 1 diabetes as well (Page 18, Lines 367-376). This shows that our findings in individuals with type 1 diabetes are comparable to the best studies in the field.

 

Reviewer #2

1. The study proposes a straight forward strategy of predicting blood glucose levels using ML models. The models are trained with a large dataset of 851 patients. The dataset contains data from T2Ds, prediabetics and normal individuals. The forecasting is done for a PH of 15 and 60 minutes. The results show almost perfect prediction, this is due to methodological errors. The authors claim to have split training, cross-validation and test data randomly. This could prove to be a wrong strategy in time-series forecasting as there is a chance of the model getting trained on the future data.

We agree that certain training strategies can cause the model to –during evaluation– recognize data on which it had been trained, which indeed would lead to near perfect prediction. However, as explained in the Methods section under dataset construction (Page 9, Lines 170-172), the datasets were split in such a way that any given participant was present in only the training, tuning, or evaluation set. As such, the models have been trained in other individuals than those who are present in the evaluation set. Therefore, it is not possible for the model to have been trained on ‘future data’. 

2. The authors trained multiple models for prediction purpose. It is seen in the performance comparison table that classical RNN performs best for 15 min PH and LSTM performs best for 60 min PH. The manuscript, however, only contains details about the LSTM model.

We agree with the reviewer’s comment that the classical RNN just outperformed all other models at a prediction horizon of 15 minutes. Considering 1) the differences between the RNN and LSTM (RMSE: 0.485 [0.481-0.490] vs. 0.482 [0.477-0.487]) at 15 minutes were negligible and not statistically significant; and 2) RNN was substantially worse than LSTM (RMSE: 0.989 [0.983-0.995] vs. 0.941 [0.937-0.945]) at 60 minutes, we decided to use the LSTM architecture for predictions at both time horizons. The details about the RNN and LSTM used in the baseline comparison are described in S1 Supporting Information (Page 2). 

3. Since, the proposed study does exactly the same what various other works have been doing for BG prediction during the past decade, no attempt at performance comparison with prior work has been made.

We believe that our work is unique in several respects. First, our study features one of the largest study populations to date. Second, the large-scale combination of CGM and accelerometry is unique. Third, we train glucose prediction models in individuals with NGM, prediabetes, or type 2 diabetes, research on which is notably scarce.

As our initial findings were obtained from a population ranging from normal glucose metabolism to type 2 diabetes, we compared our performance metrics in the type 2 diabetes subgroup with the largest study in type 2 diabetes to date (n=50)(4). This can be found in the Discussion (Page 17-18, Lines 355-367). Initially, we did not set our findings against studies in individuals with type 1 diabetes because comparison would not be valid (individuals with type 1 diabetes experience much greater daily glucose variability). As we now have extended our results to individuals with type 1 diabetes, we have also included a comparison to the most recent and best-performing studies in type 1 diabetes (Page 18, Lines 367-376).

4. Performance improvement depicted in the CGM+PA dataset is not significant and hence provides no motive for designers to prefer one over the other.

We agree with the reviewer that model performance improves only slightly when accelerometer data is incorporated, as we have delineated this in our Discussion accordingly (e.g., Page 17, Lines 350-351; Page 18, Lines 378-393). Nevertheless, our study is the first to use such a large population to make this important comparison.

5. Since it is understood that the glucose variability in NGM is low, and the number of individuals with NGM in both datasets are the largest, the underlying trends being identified by the ML model are overwhelmed by such data. It explains why the ML model are predicting almost perfectly.

We assent with the reviewer that prediction is (expected to be) most accurate in individuals with NGM. Hence, we stratified the performance metrics based on the participants’ glucose metabolism status in order to assess how accuracy would fare for participants with prediabetes or type 2 diabetes (Table 2, S5 Table). Furthermore, the clinical safety results are shown only for individuals with type 2 diabetes (Figure 2). Based on these data, we believe that the high accuracy and clinical safety in individuals with type 2 diabetes are not caused by the model being overwhelmed by data of participants with NGM. This is now further supported by the finding that our models translate quite well to individuals with type 1 diabetes. 

 

Reviewer #3 

1. This article presents the work on the application of different machine learning techniques for glucose prediction from CGM and physical activity bracelets. This is a study with a very large number of patients but in my opinion the article is not interesting for the journal for several reasons. Firstly, the data are not available to the research community, which makes it difficult to check whether the techniques presented can be overcome by the countless number of papers in the area.

Data of The Maastricht Study are certainly available to researchers who meet the criteria for access to confidential data. For the safety and privacy of the participants, as requested by law and ethical regulations, strict procedures to obtain data are in place. The implication of this is that the data have been deemed unsuitable for public deposition by The Board of The Maastricht Study, as described in detail under Data availability (Page 22, Lines 465-472). This should not be a ground to preclude publication of our work in PLOS ONE. Accordingly, multiple manuscripts that used data from The Maastricht Study and, thus, were under the same restrictions regarding data availability have been published in PLOS ONE(5-9). 

2. Secondly, no new techniques are proposed, there are many studies in the area and the techniques of machine learning have been studied in depth, the authors can see for example all the articles in results were reported in different previous publications recently and some years ago. You can see, for instance the works presented at the last two workshops on Blood Glucose Level Prediction (BGLP) Challenge http://ceur-ws.org/Vol-2675/ You can also find several journal papers

 Hidalgo, J. I., Colmenar, J. M., Kronberger, G., Winkler, S. M., Garnica, O., & Lanchares, J. (2017). Data based prediction of blood glucose concentrations using evolutionary methods. Journal of medical systems, 41(9), 142.

 Woldaregay, A. Z., Årsand, E., Walderhaug, S., Albers, D., Mamykina, L., Botsis, T., & Hartvigsen, G. (2019). Data-driven modeling and prediction of blood glucose dynamics: Machine learning applications in type 1 diabetes. Artificial intelligence in medicine, 98, 109-134.

 Velasco, J. M., Garnica, O., Lanchares, J., Botella, M., & Hidalgo, J. I. (2018). Combining data augmentation, EDAs and grammatical evolution for blood glucose forecasting. Memetic Computing, 10(3), 267-277.

 De Falco, I., Della Cioppa, A., Giugliano, A., Marcelli, A., Koutny, T., Krcma, M., ... & Tarantino, E. (2019). A genetic programming-based regression for extrapolating a blood glucose-dynamics model from interstitial glucose measurements and their first derivatives. Applied Soft Computing, 77, 316-328.

 Contreras, I., & Vehi, J. (2018). Artificial intelligence for diabetes management and decision support: literature review. Journal of medical Internet research, 20(5),

And even more from the last two years on NNs and DL approaches

We thank the reviewer for these literature suggestions. We would like to point out that such references were not included because of our focus on type 2 diabetes. Since we have now expanded our work to include translation of our prediction models to individuals with type 1 diabetes, we have updated our references accordingly. Our findings are comparable to most recent findings, as summarized in the table below (available in the reviewer document). 

In addition, we would like to point out that the purpose of our study was not to develop a completely new glucose prediction technique. On the one hand, we aimed to investigate how well a glucose prediction model would fare in a large study population with actual CGM data (in contrast to in silico data). On the other hand, we aimed to assess whether incorporation of simultaneously assessed accelerometry data would improve glucose prediction. We certainly have the ambition to develop our models further (e.g., by using more complex or novel algorithms), but view such endeavors to be out of scope for the current manuscript.

3. Moreover a prediction horizon of 15 minutes is so short that any Naive approach could reach a 95% of safe predictions, I recommend the authors the exercise of predicting the glucose value for 15 minutes as the value a t=0. 

As the reviewer suggested, we calculated the performance characteristics of a model predicting the glucose value at t0. The results are shown in S7 Table and S3 Figure. As the reviewer can appreciate, naive prediction performance is substantially worse than our ML-based prediction models, especially in the type 2 diabetes.

We further analyzed the effect of using t0 as prediction value between prediction horizons 0 and 120 minutes. S4 Figure illustrates the effect for each of the performance measures. 

4. Experimental results are not useful as they are presented in the paper. All the techniques are summarized in just one table and no discrimination among them is done.

First, we want to highlight that we have compared a large number of different machine learning models (i.e., ARIMA, support vector regression, gradient-boosting trees, feed-forward neural networks, and recurrent neural networks [RNN]), which is outlined in the Methods (Page 9, Lines 183-187). The results of the comparisons are indeed shown in one supplementary table (S1 Table). However, we do not concur that this is a drawback of our study. The aims of our study were to assess to what extent glucose values can be accurately predicted at 15- and 60-minute intervals, and whether it could be further improved by incorporation of accelerometer-measured physical activity data. We intended to find the best machine learning-based prediction model as a mains to an end, not as a goal in itself. After we concluded that LSTM performed best, especially at a prediction interval of 60 minutes, there was no need to further address or compare all other possible techniques. Still, we did further optimize the architecture of the LSTM-based models; the hyperparameters that were studied and chosen are shown in S2 and S3 Table.

5. The main conclusion of the paper is so general that is obvious. Is something like the affirmation " Medicine is good" or something similar.

We have now rewritten the main conclusion in order to incorporate our validation in type 1 diabetes. We believe that this overcomes the point made by the reviewer.

6. Last but no least, the study affirm that, although it was made with T2 diabetes patients, it could be extrapolated to other T1 patients. I am sure that conclusions for T2 can not directly extrapolated to other type of patients. It has been shown in the past that in-silico results are not extensible to T1 real patients nor to T2 and vice versa. Glucose Variability of one T1 or T2 patients are different, T1 can produce little amounts of insulin or not, T2 insulin resistance could be heavier for one patient than for other....

We agree with the reviewer that our findings cannot automatically be translated to a T1D population. Based on the reviewer’s remark, we have therefore extended our results to a small study population of individuals with type 1 diabetes (OhioT1DM Dataset). As can be appreciated from Figure 3, the clinical safety of our prediction models is indeed quite high, even at the 60-minute interval (>91% predictions are highly clinically safe). Furthermore, the accuracy of our models is comparable to current studies in type 1 diabetes. We have rewritten our discussion to include our reflection on the findings for type 1 diabetes (Page 17, Lines 367-376).

7. So in this conditions the study is of little interest for the journal, In my humble opinion, I think that the data set has a great potential and that the research team is capable of prepare and in depth analysis of machine learning technique, I would recommend to separate and configure ML techniques for the different types of patients, and of course when presenting the results separate by ML techniques and data sets.

We would like to thank the reviewer for acknowledging the great potential of our dataset. Still, we want to reiterate what the main purposes of our study were. First, to investigate to what extent glucose values can be accurately predicted at 15- and 60-minute intervals, while using the best performing ML-based model in a large sample of participants with either NGM, prediabetes, or type 2 diabetes. Second, whether prediction could be further improved by incorporation of accelerometer-measured physical activity data. Finally, we believe that the (clinical) interest has been augmented by inclusion of a T1D dataset in the revised version of the manuscript, for which the reviewer is greatly acknowledged. 

Reviewer #4 

1. In this paper, the authors used machine learning to train models in predicting future glucose levels based on prior CGM and accelerometry data. According to experiments the authors conducted, the results showed that machine learning-based models are able to accurately and safely predict glucose values both at 15- and 60-minute intervals with only CGM data; And incorporation of accelerometer data slightly improved prediction. It is interesting and of great value to utilize machine learning-based models to predict future glucose levels. At the same time, I have several major concerns about this study. First, the authors did not conduct a literature review of researches on future glucose level predictions. Based on this manuscript, we do not know which models are used to predict future glucose levels, and how the current progress is, especially the use of machine learning-based models that this paper employed. Further, it is also difficult to determine whether this paper has enough innovations and contributions, as the authors did not list in detail.

In the revised Discussion of our manuscript, we now extensively compare our results with literature results (Page 17-18, Lines 355-376). We also refer to a recent review paper on this topic in the Introduction section (Page 5, Lines 87-89)(13). Of note, the true purposes of our study were to assess to what extent glucose values can be accurately predicted at 15- and 60-minute intervals in individuals with NGM, prediabetes, or type 2 diabetes, and whether the prediction could be further improved by incorporation of accelerometer-measured physical activity data. Unique in this regard are our large study population of individuals with NGM, prediabetes, or type 2 diabetes (n=851) and the large-scale combination of simultaneously performed CGM and accelerometry (n=540).

2. Second, in the “Model development and design” part of the article, the authors mentioned that the prediction task of future glucose levels can be solved by a variety of statistical and machine learning models, and the authors assessed various models, such as autoregressive integrated moving average, support vector regression, etc. Finally, a LSTM architecture was chosen as this had the best performance in the tuning dataset. There are two questions of the model selection: 

1) Why the authors selected these statistical and machine learning models? What are the applicability and advantages of these models?

The statistical and machine learning models used were selected on the basis of previously published literature in relation to time-series forecasting and glucose prediction. Details and applicability for each of these models were briefly discussed in S1 Supporting Information (Page 2, Lines 24-56). 

2) The authors utilized the multi-task LSTM network to predict future glucose levels finally as it had the best performance in the tuning dataset. If the dataset is modified or other features like diet, medication use are added in the models, whether the multi-task LSTM network can also achieve the best performance is unknown. Therefore, except for the reason of best performance, the authors should detail other reasons for choosing the multi-task LSTM network.

Our major rationale to select the LSTM architecture is the baseline performance described in S1 Supporting Information (Page 2, Lines 24-56) and S1 Table. Additional advantages include the option to incorporate explainability into our LSTM models (14, 15), and the relatively low computing cost compared to complex, deep neural networks which could potentially hinder application of these networks in relatively simple devices such as closed-loop insulin delivery systems. Nonetheless, we agree with the author that in case additional features, such as diet or medication use, would be added to the dataset, we would have to reevaluate our current architecture. We have included this in our Limitations section (Page 20, Lines 430-431).

3. Third, the authors verified the performance of the multi-task LSTM network with 851 individuals. As is known to all, deep learning models are usually validated on a large number of datasets. The amount of data in this paper may not be convincing enough for model validation. 

We agree with the reviewer that deep learning models should be validated on a large number of samples in order to confidently assess its generalizability. In our specific study, it is important to realize that, besides the number of individuals, the number of glucose measurements are also critical in the training of these models. We had almost 1.4 million glucose measurements available in the our study. Currently, this is the largest study that described the application of ML-based glucose prediction in individuals with type 2 diabetes. Nevertheless, we acknowledge that future studies should assess to what extent our models generalize to other populations. 

References

1. Kumareswaran K, Thabit H, Leelarathna L, Caldwell K, Elleri D, Allen JM, et al. Feasibility of closed-loop insulin delivery in type 2 diabetes: a randomized controlled study. Diabetes Care. 2014;37(5):1198-203.

2. Li K, Liu C, Zhu T, Herrero P, Georgiou P. GluNet: A Deep Learning Framework for Accurate Glucose Forecasting. IEEE J Biomed Health Inform. 2020;24(2):414-23.

3. Perez-Gandia C, Facchinetti A, Sparacino G, Cobelli C, Gomez EJ, Rigla M, et al. Artificial neural network algorithm for online glucose prediction from continuous glucose monitoring. Diabetes Technol Ther. 2010;12(1):81-8.

4. Mohebbi A, Johansen AR, Hansen N, Christensen PE, Tarp JM, Jensen ML, et al. Short Term Blood Glucose Prediction based on Continuous Glucose Monitoring Data. arXiv e-prints [Internet]. 2020 February 01, 2020:[arXiv:2002.02805 p.]. Available from: https://ui.adsabs.harvard.edu/abs/2020arXiv200202805M.

5. de Rooij BH, van der Berg JD, van der Kallen CJ, Schram MT, Savelberg HH, Schaper NC, et al. Physical Activity and Sedentary Behavior in Metabolically Healthy versus Unhealthy Obese and Non-Obese Individuals - The Maastricht Study. PLoS One. 2016;11(5):e0154358.

6. Sorensen BM, Houben A, Berendschot T, Schouten J, Kroon AA, van der Kallen CJH, et al. Cardiovascular risk factors as determinants of retinal and skin microvascular function: The Maastricht Study. PLoS One. 2017;12(10):e0187324.

7. Elissen AMJ, Hertroijs DFL, Schaper NC, Bosma H, Dagnelie PC, Henry RM, et al. Differences in biopsychosocial profiles of diabetes patients by level of glycaemic control and health-related quality of life: The Maastricht Study. PLoS One. 2017;12(7):e0182053.

8. Martens RJH, van der Berg JD, Stehouwer CDA, Henry RMA, Bosma H, Dagnelie PC, et al. Amount and pattern of physical activity and sedentary behavior are associated with kidney function and kidney damage: The Maastricht Study. PLoS One. 2018;13(4):e0195306.

9. Consolazio D, Koster A, Sarti S, Schram MT, Stehouwer CDA, Timmermans EJ, et al. Neighbourhood property value and type 2 diabetes mellitus in the Maastricht study: A multilevel study. PLoS One. 2020;15(6):e0234324.

10. Kriventsov S, Lindsey A, Hayeri A. The Diabits App for Smartphone-Assisted Predictive Monitoring of Glycemia in Patients With Diabetes: Retrospective Observational Study. JMIR Diabetes. 2020;5(3):e18660.

11. Martinsson J, Schliep A, Eliasson B, Mogren O. Blood Glucose Prediction with Variance Estimation Using Recurrent Neural Networks. Journal of Healthcare Informatics Research. 2020;4(1):1-18.

12. Chen J, Li K, Herrero P, Zhu T, Georgiou P, editors. Dilated Recurrent Neural Network for Short-time Prediction of Glucose Concentration. KHD@IJCAI; 2018.

13. Woldaregay AZ, Arsand E, Walderhaug S, Albers D, Mamykina L, Botsis T, et al. Data-driven modeling and prediction of blood glucose dynamics: Machine learning applications in type 1 diabetes. Artif Intell Med. 2019;98:109-34.

14. Thorsen-Meyer H-C, Nielsen AB, Nielsen AP, Kaas-Hansen BS, Toft P, Schierbeck J, et al. Dynamic and explainable machine learning prediction of mortality in patients in the intensive care unit: a retrospective study of high-frequency data in electronic patient records. The Lancet Digital Health. 2020.

15. Lauritsen SM, Kristensen M, Olsen MV, Larsen MS, Lauritsen KM, Jorgensen MJ, et al. Explainable artificial intelligence model to predict acute critical illness from electronic health records. Nat Commun. 2020;11(1):3852.

---

## [Decision Letter · Decision Letter 1]

19 Mar 2021

PONE-D-20-30681R1

Machine learning-based glucose prediction with use of continuous glucose and physical activity monitoring data: The Maastricht Study

PLOS ONE

Dear Dr. Brouwers,

Thank you for submitting your manuscript to PLOS ONE. After careful consideration, we feel that it has merit but does not fully meet PLOS ONE’s publication criteria as it currently stands. Therefore, we invite you to submit a revised version of the manuscript that addresses the points raised during the review process.

We look forward to receiving your revised manuscript.

Kind regards,

Chi-Hua Chen, Ph.D.

Academic Editor

PLOS ONE

Reviewers' comments:

Reviewer's Responses to Questions

**Comments to the Author**

1. If the authors have adequately addressed your comments raised in a previous round of review and you feel that this manuscript is now acceptable for publication, you may indicate that here to bypass the “Comments to the Author” section, enter your conflict of interest statement in the “Confidential to Editor” section, and submit your "Accept" recommendation.

Reviewer #1: All comments have been addressed

Reviewer #3: (No Response)

2. Is the manuscript technically sound, and do the data support the conclusions?

Reviewer #1: Yes

Reviewer #3: No

3. Has the statistical analysis been performed appropriately and rigorously? 

Reviewer #1: Yes

Reviewer #3: No

4. Have the authors made all data underlying the findings in their manuscript fully available?

Reviewer #1: Yes

Reviewer #3: No

5. Is the manuscript presented in an intelligible fashion and written in standard English?

Reviewer #1: Yes

Reviewer #3: Yes

6. Review Comments to the Author

Reviewer #1: The paper has been improved significantly. All my comments have been addressed with clear explanation and all updates have been shown explicitly in the paper.

Reviewer #3: I think that my concerns have not been addressed.

The paper has little interest for the reader of the journal. In the case of people not working in the field, the contribution is so poor that no extrapolation to other works can be done. On the other hand for people working on this problem, conclusion are known, statistical validation is not made and conclusions are not fully supported by experiments.

The inclusion of T1D patients is forced in my opinion and does not make much sense with the other results.

My questions are again the same

2. Secondly, no new techniques are proposed,....

Analyses are mere description of the results, see for instance lines 323 to 329:

323 Additional analyses

324 To further obtain insights into our model predictions, we assessed performance metrics

325 stratified by day and night (S8 Table). Fifteen-minute predictions did not materially differ

326 between day and night. By contrast, accuracy of 60-minute predictions was lower during the

327 day than at night. In addition, we stratified the results by high or low glucose variability (i.e.,

328 SD cut-off of 1.37 mmol/L) (S9 Table). Model performance was slightly lower at higher

329 glucose variability, at both time intervals of 15 and 60 minutes.

4. Experimental results are not useful as they are presented in the paper. All the techniques are summarized in just one table and no discrimination among them is done.

7. PLOS authors have the option to publish the peer review history of their article (what does this mean?). If published, this will include your full peer review and any attached files.

Reviewer #1: No

Reviewer #3: No

---

## [Author Response · Author response to Decision Letter 1]

9 Apr 2021

We would like to thank the reviewers for their feedback on our study and for the time spent on our manuscript. Please find below our point-by-point rebuttal.

Reviewer #1

The paper has been improved significantly. All my comments have been addressed with clear explanation and all updates have been shown explicitly in the paper.

We once more would like to thank the reviewer for his/her positive feedback on our study and for the time spent on our manuscript.

Reviewer #3 

I think that my concerns have not been addressed. The paper has little interest for the reader of the journal. In the case of people not working in the field, the contribution is so poor that no extrapolation to other works can be done. On the other hand for people working on this problem, conclusion are known, statistical validation is not made and conclusions are not fully supported by experiments.

We respectfully disagree with the comments made by the reviewer. We would like to reiterate the main aims of our study. First, we investigated to what extent glucose values can be accurately predicted at 15- and 60-minute intervals, while using the best performing ML-based model in a large sample of participants with either normal glucose metabolism (NGM), prediabetes, or type 2 diabetes. Such large scale data has not yet been used in the context of a glucose prediction study. Second, we assessed whether prediction could be further improved by incorporation of accelerometer-measured physical activity data. Such a large-scale combination of simultaneously collected continuous glucose monitoring and activity tracker has not yet been used. Finally, we augmented the (clinical) interest of our study by including model validation in a type 1 diabetes dataset in the revised version of the manuscript, for which the reviewer is greatly acknowledged. In conclusion, the combination of our large cohort of individuals with NGM, prediabetes or type 2 diabetes as well as the examination of glucose measurements combined with physical activity data has never been described to date and can, therefore, be of great interest to the readers of PLOS ONE. 

The inclusion of T1D patients is forced in my opinion and does not make much sense with the other results.

We chose to include individuals with type 1 diabetes in order to examine the performance of our models in this subgroup as part of a proof-of-concept analysis. As described in our Discussion, we agree that further, comprehensive evaluation is necessary in order to examine the real clinical benefit and performance of our models in individuals with type 1 diabetes. Notably, previous studies have also employed this dataset in order to validate their models (1-7).

 

My questions are again the same 2. Secondly, no new techniques are proposed,....

Analyses are mere description of the results, see for instance lines 323 to 329:

Additional analyses

To further obtain insights into our model predictions, we assessed performance metrics stratified by day and night (S8 Table). Fifteen-minute predictions did not materially differ between day and night. By contrast, accuracy of 60-minute predictions was lower during the day than at night. In addition, we stratified the results by high or low glucose variability (i.e., SD cut-off of 1.37 mmol/L) (S9 Table). Model performance was slightly lower at higher glucose variability, at both time intervals of 15 and 60 minutes.

We again would like to stress that the main objective of the current work is not the technological advancement of algorithms per se. We certainly have the ambition to develop our models further (e.g., by using more complex or innovative algorithms), but view such endeavors to be out of scope for the current manuscript.

4. Experimental results are not useful as they are presented in the paper. All the techniques are summarized in just one table and no discrimination among them is done.

We assume the reviewer is referencing to the baseline comparison of various statistical and machine learning models as depicted in S1 Table. In line with our previous comment, we would like to point out that we did not aim to achieve technological advancements. Hence, we intended to find the best machine learning-based prediction model as a means to an end, not as a goal in itself. After we concluded that LSTM performed best, especially at a prediction interval of 60 minutes, there was no need to further address or compare all other possible techniques. As such, the comprehensive baseline evaluation of different statistical and machine learning models can even be considered a strength of our study. 

 

References

1. Marling C, Bunescu RC, editors. The OhioT1DM Dataset For Blood Glucose Level Prediction. KHD@IJCAI; 2018.

2. Marling C, Bunescu RC, editors. The OhioT1DM Dataset For Blood Glucose Level Prediction: Update 2020. KHD@IJCAI; 2020.

3. Pappada SM, Owais MH, Cameron BD, Jaume JC, Mavarez-Martinez A, Tripathi RS, et al. An Artificial Neural Network-based Predictive Model to Support Optimization of Inpatient Glycemic Control. Diabetes Technol Ther. 2020;22(5):383-94.

4. Li K, Liu C, Zhu T, Herrero P, Georgiou P. GluNet: A Deep Learning Framework for Accurate Glucose Forecasting. IEEE J Biomed Health Inform. 2020;24(2):414-23.

5. Chen J, Li K, Herrero P, Zhu T, Georgiou P, editors. Dilated Recurrent Neural Network for Short-time Prediction of Glucose Concentration. KHD@IJCAI; 2018.

6. Martinsson J, Schliep A, Eliasson B, Meijner C, Persson S, Mogren O. Automatic blood glucose prediction with confidence using recurrent neural networks. 3rd International Workshop on Knowledge Discovery in Healthcare Data, KDH@IJCAI-ECAI 2018, 13 July 2018; 20182018. p. 64-8.

7. Kriventsov S, Lindsey A, Hayeri A. The Diabits App for Smartphone-Assisted Predictive Monitoring of Glycemia in Patients With Diabetes: Retrospective Observational Study. JMIR Diabetes. 2020;5(3):e18660.

---

## [Decision Letter · Decision Letter 2]

23 Apr 2021

PONE-D-20-30681R2

Machine learning-based glucose prediction with use of continuous glucose and physical activity monitoring data: The Maastricht Study

PLOS ONE

Dear Dr. Brouwers,

Thank you for submitting your manuscript to PLOS ONE. After careful consideration, we feel that it has merit but does not fully meet PLOS ONE’s publication criteria as it currently stands. Therefore, we invite you to submit a revised version of the manuscript that addresses the points raised during the review process.

We look forward to receiving your revised manuscript.

Kind regards,

Chi-Hua Chen, Ph.D.

Academic Editor

PLOS ONE

Journal Requirements:

Reviewers' comments:

Reviewer's Responses to Questions

**Comments to the Author**

1. If the authors have adequately addressed your comments raised in a previous round of review and you feel that this manuscript is now acceptable for publication, you may indicate that here to bypass the “Comments to the Author” section, enter your conflict of interest statement in the “Confidential to Editor” section, and submit your "Accept" recommendation.

Reviewer #1: All comments have been addressed

Reviewer #3: (No Response)

2. Is the manuscript technically sound, and do the data support the conclusions?

Reviewer #1: Yes

Reviewer #3: Yes

3. Has the statistical analysis been performed appropriately and rigorously? 

Reviewer #1: Yes

Reviewer #3: Yes

4. Have the authors made all data underlying the findings in their manuscript fully available?

Reviewer #1: No

Reviewer #3: No

5. Is the manuscript presented in an intelligible fashion and written in standard English?

Reviewer #1: Yes

Reviewer #3: Yes

6. Review Comments to the Author

Reviewer #1: My all comments have been addressed properly. The only issue is that, Plus One requires all data underlying the findings in the manuscript fully available. It is better to address this issue before publish.

Reviewer #3: I really appreciate the efforts of the authors for improving the paper and answering my questions. I would like to explain better which is my opinion about the great potential of the work.

What I would expect of such amount of data is to obtain guidelines for selecting and designing better ML (or not ML) algorithms, based on the precision needed, the time for response, the data availability and of course the features of the patient.

For me, what it would be useful for the journal readers is a combination of tables 1 and 2 with information provided as supporting information.

As the supporting information is going to be publish, a summary of this information should be included in the paper, in order to highlight the insights of this study. I would like to see a Table with a summary of the supporting information in the main paper

The paper is is a very interesting work

7. PLOS authors have the option to publish the peer review history of their article (what does this mean?). If published, this will include your full peer review and any attached files.

Reviewer #1: No

Reviewer #3: No

---

## [Author Response · Author response to Decision Letter 2]

20 May 2021

We would like to thank the reviewers for their feedback on our study and for the time spent on our manuscript. Please find below our point-by-point rebuttal. The page and line numbers refer to the manuscript version with track changes.

Reviewer #1

My all comments have been addressed properly. The only issue is that, Plus One requires all data underlying the findings in the manuscript fully available. It is better to address this issue before publish.

Data of The Maastricht Study are certainly available to researchers who meet the criteria for access to confidential data. For the safety and privacy of the participants, as requested by law and ethical regulations, strict procedures to obtain data are in place. The implication of this is that the data have been deemed unsuitable for public deposition by The Board of The Maastricht Study, as described in detail under Data availability (Page 24, Lines 493-498). This should not be a ground to preclude publication of our work in PLOS ONE. Accordingly, multiple manuscripts that used data from The Maastricht Study and, thus, were under the same restrictions regarding data availability have been published in PLOS ONE(1-5).

Reviewer #3

I really appreciate the efforts of the authors for improving the paper and answering my questions. I would like to explain better which is my opinion about the great potential of the work. What I would expect of such amount of data is to obtain guidelines for selecting and designing better ML (or not ML) algorithms, based on the precision needed, the time for response, the data availability and of course the features of the patient. For me, what it would be useful for the journal readers is a combination of tables 1 and 2 with information provided as supporting information. As the supporting information is going to be publish, a summary of this information should be included in the paper, in order to highlight the insights of this study. I would like to see a Table with a summary of the supporting information in the main paper. The paper is is a very interesting work

As suggested by the reviewer, we have now included the baseline comparison of statistical and machine learning models for glucose prediction in our main manuscript as Table 1 (Page 12, Lines 223-225). Accordingly, we made various changes in the methods section of our main manuscript (Page 10, Lines 195-205; Page 22, Lines 439-441) and supplementary information (Page 4, Lines 92-98). 

References

1. de Rooij BH, van der Berg JD, van der Kallen CJ, Schram MT, Savelberg HH, Schaper NC, et al. Physical Activity and Sedentary Behavior in Metabolically Healthy versus Unhealthy Obese and Non-Obese Individuals - The Maastricht Study. PLoS One. 2016;11(5):e0154358.

2. Sorensen BM, Houben A, Berendschot T, Schouten J, Kroon AA, van der Kallen CJH, et al. Cardiovascular risk factors as determinants of retinal and skin microvascular function: The Maastricht Study. PLoS One. 2017;12(10):e0187324.

3. Elissen AMJ, Hertroijs DFL, Schaper NC, Bosma H, Dagnelie PC, Henry RM, et al. Differences in biopsychosocial profiles of diabetes patients by level of glycaemic control and health-related quality of life: The Maastricht Study. PLoS One. 2017;12(7):e0182053.

4. Martens RJH, van der Berg JD, Stehouwer CDA, Henry RMA, Bosma H, Dagnelie PC, et al. Amount and pattern of physical activity and sedentary behavior are associated with kidney function and kidney damage: The Maastricht Study. PLoS One. 2018;13(4):e0195306.

5. Consolazio D, Koster A, Sarti S, Schram MT, Stehouwer CDA, Timmermans EJ, et al. Neighbourhood property value and type 2 diabetes mellitus in the Maastricht study: A multilevel study. PLoS One. 2020;15(6):e0234324.

---

## [Decision Letter · Decision Letter 3]

1 Jun 2021

Machine learning-based glucose prediction with use of continuous glucose and physical activity monitoring data: The Maastricht Study

PONE-D-20-30681R3

Dear Dr. Brouwers,

We’re pleased to inform you that your manuscript has been judged scientifically suitable for publication and will be formally accepted for publication once it meets all outstanding technical requirements.

Kind regards,

Chi-Hua Chen, Ph.D.

Academic Editor

PLOS ONE

Additional Editor Comments (optional):

Reviewers' comments:

Reviewer's Responses to Questions

**Comments to the Author**

1. If the authors have adequately addressed your comments raised in a previous round of review and you feel that this manuscript is now acceptable for publication, you may indicate that here to bypass the “Comments to the Author” section, enter your conflict of interest statement in the “Confidential to Editor” section, and submit your "Accept" recommendation.

Reviewer #1: All comments have been addressed

Reviewer #3: All comments have been addressed

2. Is the manuscript technically sound, and do the data support the conclusions?

Reviewer #1: Yes

Reviewer #3: Yes

3. Has the statistical analysis been performed appropriately and rigorously? 

Reviewer #1: Yes

Reviewer #3: Yes

4. Have the authors made all data underlying the findings in their manuscript fully available?

Reviewer #1: Yes

Reviewer #3: No

5. Is the manuscript presented in an intelligible fashion and written in standard English?

Reviewer #1: Yes

Reviewer #3: Yes

6. Review Comments to the Author

Reviewer #1: All comments have been nicely addressed. Data of The Maastricht Study are certainly available to researchers who meet the criteria for access to confidential data

Reviewer #3: All my comments have been addressed. I really appreciate the efforts made to include the tables I suggested.

7. PLOS authors have the option to publish the peer review history of their article (what does this mean?). If published, this will include your full peer review and any attached files.

Reviewer #1: No

Reviewer #3: No

---

## [Editor Report · Acceptance letter]

15 Jun 2021

PONE-D-20-30681R3 

Machine learning-based glucose prediction with use of continuous glucose and physical activity monitoring data: The Maastricht Study 

Dear Dr. Brouwers:

I'm pleased to inform you that your manuscript has been deemed suitable for publication in PLOS ONE. Congratulations! Your manuscript is now with our production department. 

Kind regards, 

on behalf of

Professor Chi-Hua Chen 

Academic Editor

PLOS ONE